# Adjustment of light-responsive NADP dynamics in chloroplasts by stromal pH

Yusuke Fukuda[1,4], Chinami Ishiyama[1,4], Maki Kawai-Yamada[2] &
Shin-nosuke Hashida [3] ✉

Cyclic electron transfer (CET) predominates when $NADP^+$ is at basal levels, early in photosynthetic induction; however, the mechanism underlying the subsequent supply of $NADP^+$ to fully drive steady-state linear electron transfer remains unclear. Here, we investigated whether CET is involved in de novo $NADP^+$ supply in *Arabidopsis thaliana* and measured chloroplastic NADP dynamics to evaluate responsiveness to variable light, photochemical inhibitors, darkness, and CET activity. The sum of oxidized and reduced forms shows that levels of NADP and NAD increase and decrease, respectively, in response to light; levels of NADP and NAD decrease and increase in the dark, respectively. Moreover, consistent with the pH change in the stroma, the pH preference of chloroplast $NAD^+$ phosphorylation and $NADP^+$ dephosphorylation is alkaline and weakly acidic, respectively. Furthermore, CET is correlated with upregulation of light-responsive NADP level increases and downregulation of dark-responsive NADP level reductions. These findings are consistent with CET helping to regulate NADP pool size via stromal pH regulation under fluctuating light conditions.

When light is absorbed by plant leaves, two photosynthetic protein complexes, photosystem II (PSII) and photosystem I (PSI), are activated; PSII oxidizes water and PSI reduces $NADP^+$, via a process called linear electron transfer (LET). In PSI, electrons are transported from plastocyanin (PC) to ferredoxin (Fd), which is then passed on to $NADP^+$ by Fd:$NADP^+$ reductase to produce NADPH. During the light phase, protons are transported into the thylakoid lumen from the stroma[1,2], resulting in an increase in the stromal pH from 7.8 to 8.0, whereas the luminal pH drops below 6.0 (i.e., the pH gradient: ΔpH)[3]. In addition to LET, two different routes have been proposed to function as alternative electron acceptors from PSI. One of these routes is dependent on the NAD(P)H plastoquinone (PQ) oxidoreductase-like (NDH) complex[4,5] and the other is dependent on proton gradient regulation 5 (PGR5)/PRG5-like photosynthetic phenotype 1 (PGRL1)[6,7]. Instead of electron transfer from the reduced form of Fd to $NADP^+$ at PSI, these routes reflux electrons to PQ via cyclic electron transfer (CET) to form

ΔpH and ATP without generating NADPH. These two distinct pathways are not only redundant but the importance of the NDH pathway in stress resistance has also been reported in various plant species[8–11]. Unlike the mutant in the NDH pathway, the mutant in the PGR5 pathway demonstrated severe disturbance of photosynthetic regulation and could not induce pH-dependent non-photochemical quenching[6]. According to Kawashima et al., the contribution of the PGR5 and NDH pathways to ΔpH formation was experimentally estimated at 30% and 5%, respectively[12]. Although the critical function of both the complexes in CET remains under debate, studies have suggested that PGR5/PGRL1 is involved in LET function through the regulation of the cytochrome (Cyt) $b_6f$ complex and ATP synthetase complex under moderate light conditions in an Antimycin A-sensitive manner[13–15].

Proper distribution of electron transfer between LET and CET is important during photosynthetic induction because the NADP pool at the beginning of photosynthesis is considerably lower than that during

[1]Civil Engineering Research & Environmental Studies (CERES), Inc., 1646, Abiko, Chiba 270-1194, Japan. [2]Graduate School of Science and Engineering, Saitama University, 255 Shimo-Okubo, Sakura-ku, Saitama 338-8570, Japan. [3]Sustainable System Research Laboratory, Central Research Institute of Electric Power Industry (CRIEPI), 1646, Abiko, Chiba 270-1194, Japan. [4]These authors contributed equally: Yusuke Fukuda, Chinami Ishiyama. ✉e-mail: shashida@criepi.denken.or.jp

steady-state photosynthesis[16,17]. NADPH levels are minimal in chloroplasts in the dark (because photosynthesis does not occur). During this time, although NADP$^+$ also does not accumulate in the chloroplasts, it is present at minimal concentrations[17]. It is unclear whether this minimal amount of NADP$^+$ is sufficient to drive the LET at the onset of photosynthetic electron transfer. Photosynthetic electron transfer in PSI is limited without de novo NADP$^+$ supply even in low light[18]. CET, but not LET, has been reported to demonstrate maximal operation, after acclimating to the dark[19]; however, it remains unclear what triggers the switch in the electron-transfer pathway. Considering that the bifurcation point is the flow path from Fd, controlling de novo NADP$^+$ supply is likely important.

NAD$^+$ is the only identified precursor in the synthesis of NADP$^{+20–22}$. The final step of NAD$^+$ biosynthesis occurs in the cytoplasm, after which NAD$^+$ is distributed to each organelle; subsequently, NADP$^+$ is synthesized by ATP-dependent NAD kinase (NADK)[23]. In *Arabidopsis*, cytosolic and chloroplastic NADP$^+$ are produced by NADK1 and NADK2, respectively[24,25]. In the peroxisome, NADK3 produces NADPH from NADH[26]. Hence, NADP synthesis is thought to be independently and spatiotemporally regulated according to the demand of each cell organelle, and the sum of NADP$^+$ and NADPH (i.e., NADP pool size) varies depending on environmental and/or plant developmental conditions. Light-induced conversion of NAD$^+$ to NADP$^+$ was first discovered in *Chlorella* in 1959[27]. Similar phenomena were reported in higher plant leaves and the relationship with photosynthesis has been studied in *Chlorella*[28–30]. However, the regulatory mechanism involved in NADP dynamics remains unclear. Recently, characterization of the *nadk2* mutant lacking chloroplastic NAD$^+$ kinase revealed that light conditions may regulate NADP$^+$ production in chloroplasts[17].

In this study, we hypothesized that CET is involved in de novo NADP$^+$ supply, and we tested this hypothesis by examining the effects of electron-transfer inhibition, CET deficiency, and CET progression on NADP dynamics. We found that the correlation of CET in regulating the increase of NADP pool size is attributed to ΔpH formation. Recent advances in NADP$^+$ measurement using luminescence-based assays have improved the detection limit, quantitative reproducibility, and reliability, enabling accurate detection of the response to changes in light intensity[31]. Furthermore, we found that ΔpH resolution contributes to a decrease in NADP pool size under dark conditions.

## Results

### Light increases the NADP pool in chloroplasts by activating NAD$^+$ phosphorylation

In this study, we determined that NADP pool size was at basal levels before light exposure and immediately increased in response to illumination, reaching a plateau within 30 min (Fig. 1a). By contrast, NAD pool size decreased with increasing NADP pool size (Fig. 1b). Light intensity substantially influenced photochemistry and NADPH-generating kinetics; we found that a trace-light intensity (10 μmol m$^{-2}$ s$^{-1}$) resulted in inefficient NADPH generation (Supplementary Fig. 1). In contrast, NADP$^+$ kinetics were similar across different light intensities (Supplementary Fig. 1). Furthermore, the size of the NADP pool decreased when light intensity was further lowered from 40 (low-light intensity) to 10 μmol m$^{-2}$ s$^{-1}$ (trace-light intensity) (Fig. 1c, $p < 0.001$ one-way ANOVA), suggesting the importance of chloroplastic NADP pool size in regulating photosynthetic activity and yield based on light intensity.

A previous report demonstrated that NADK2 is activated under light conditions[17]. In the present study, we investigated the in vivo properties of chloroplast-specific NADP synthesis using isolated intact chloroplasts to evaluate increases in NADP pool size in response to light. Our results demonstrated that NADP pool size decreased by 0.14 nmol mg$^{-1}$ and 0.47 nmol mg$^{-1}$ chlorophyll (Chl) under trace-light (10 μmol m$^{-2}$ s$^{-1}$) and low-light (40 μmol m$^{-2}$ s$^{-1}$) conditions, respectively,

when intact chloroplasts suspended in isolation buffer were exposed to light for 15 min [Fig. 1d, NAD$^+$ (-)]. In contrast, the addition of NAD$^+$ to the external solution increased NADP pool size along with a net increase of 0.47 nmol mg$^{-1}$ and 0.069 nmol mg$^{-1}$ Chl under trace-light and low-light conditions [Fig. 1d, NAD$^+$ (+)], respectively. Additionally, the gross increase in NADP pool size was estimated at 0.61 nmol mg$^{-1}$ and 0.54 nmol mg$^{-1}$ Chl under trace-light and low-light conditions, respectively. Based on the gross NADP increase, NADP synthesis activity was estimated at 5.0 pmol min$^{-1}$ mg$^{-1}$, 45.7 pmol min$^{-1}$ mg$^{-1}$, and 40.8 pmol min$^{-1}$ mg$^{-1}$ Chl under dark, trace-light, and low-light conditions, respectively (Fig. 1e). Thus, consistent with the results of NADP dynamics in leaf discs (Fig. 1a), we detected light-driven NADP$^+$ synthesis in isolated chloroplasts. The in vitro NADK activity of the ruptured chloroplasts demonstrated clear pH specificity and was activated under alkaline conditions in proportion to light intensity (Fig. 1f and Supplementary Fig. 2).

### CET-dependent ΔpH formation correlates with NADP$^+$ production in chloroplasts

In chloroplasts, pH-dependent protein modification is an important process, and light-driven ΔpH formation is essential for photosynthetic regulation[32]. As photochemical inhibitors, 3-(3,4-dichlorophenyl)−1,1-dimethylurea (DCMU) and 2,5-dibromo-3-methyl-6-isopropyl-p-benzoquinone (DBMIB), prevent PSII electron transfer and Cyt $b_6f$ activity, they efficiently dissipated ΔpH formation (Supplementary Fig. 3). Moreover, they inhibited NADPH generation in LET (Supplementary Fig. 4) as well as impaired light-driven increase in NADP pool size in a dose-dependent manner (Fig. 2a, $p < 0.001$ two-way ANOVA). This inhibitory effect was further confirmed in isolated intact chloroplasts (Fig. 2b, $p < 0.05$ Tukey's HSD test). We observed a similar inhibitory effect on NADP pool size with the use of ammonium chloride, which inhibits proton pump activity during photochemical electron flow[33,34] in leaf disc (Fig. 2c and Supplementary Fig. 4, $p < 0.001$ two-way ANOVA) and in isolated intact chloroplasts (Fig. 2d, $p < 0.01$ Tukey's HSD test). Furthermore, the impairment of the light-driven NADP increase was confirmed using Nigericin, a monovalent cation-transporting ionophore that acts as an electroneutral antiporter that equilibrates K$^+$ and H$^+$ across the membrane, dissipating ΔpH, but preserving membrane potential (Supplementary Fig. 4e, f). Pretreatment with Antimycin A, an efficient and specific inhibitor of the PGR5/PGRL1 pathway of CET, also inhibited the increase in light-driven NADP pool size in chloroplasts (Fig. 2d, $p < 0.01$ Tukey's HSD test and Supplementary Fig. 5).

A previous study reported high CET activity at the beginning of light exposure[35], suggesting that CET contributes to initial ΔpH formation, which may promote de novo NADP$^+$ synthesis by stromal alkalinization as shown in Fig. 1f. Indeed, *pgr5* demonstrated clear retardation of increase in the NADP pool despite comparable NADPH generation (Fig. 2e, $p = 0.038$ $t$-test at 5 min after light on, Supplementary Fig. 6). In contrast, mutants of the *NDH* pathway [*chlororespiratory reduction* (*crr*)*2-2* and *crr4-2*] showed a similar increase in the NADP pool along with comparable kinetics for NADP$^+$ and NADPH (Fig. 2f, $p = 0.373$ and $p = 0.941$ $t$-test at 5 min after light on, Supplementary Fig. 6). These results indicate a greater contribution of the PGR5/PGRL1 pathway in increasing NADP pool size compared to the NDH pathway, possibly owing to its greater contribution toward ΔpH formation than the NDH pathway[12].

We evaluated NADK activity in vitro to investigate whether NADK activity is directly involved in the inhibition of light-responsive increase in NADP pool size (Fig. 2a, e). Both DCMU and DBMIB inhibited the enhancement of NADK activity ($p < 0.001$ one-way ANOVA), and co-treatment with DTT abolished this inhibition (Supplementary Fig. 7a, $p = 0.344$ one-way ANOVA). Therefore, redox regulation downstream of the photosystem may be involved in the process of NADK activation. Meanwhile, there was no change in the in vitro NADK

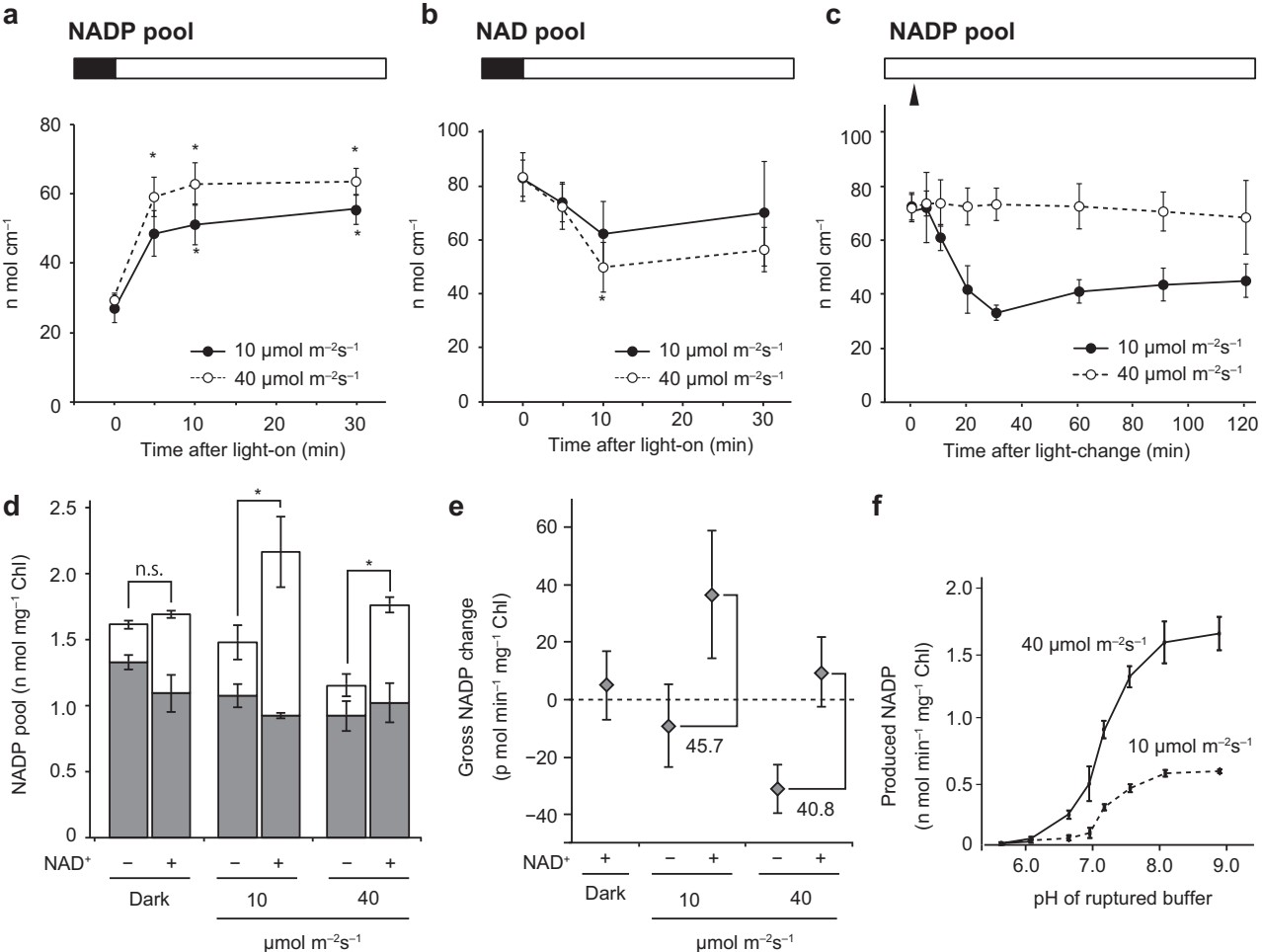

**Fig. 1 | Light-dependent production of NADP+ by NAD+ phosphorylation in chloroplasts. a**, **b** Response to light exposure: NADP (**a**) and NAD (**b**) pools in 2 h dark-acclimated *Arabidopsis* leaves. The black bar at the top of the graph indicates dark conditions and the white bar indicates light conditions. Asterisks indicate statistically significant differences from NADP or NAD pool size in the dark ($p < 0.05$, Dunnett's test). Data represent mean ± SD ($n = 6$ biological replicates). **c** Light-intensity dependence of NADP in *Arabidopsis* leaves light-acclimated for 2 h. The white bar at the top of the graph indicates the light condition, indicating that the light intensity was readjusted at the triangular point after acclimatization at 40 μmol m$^{-2}$ s$^{-1}$. Data represent mean ± SD ($n = 6$ biological replicates). **d** Light-

dependent changes in NADP+ (gray bar) and NADPH (white bar) levels in isolated chloroplasts 15 min after illumination at the indicated light intensity in the presence (+) or absence (−) of NAD+ ($n = 3$ biological replicates). Asterisks indicate statistically significant differences between the presence and absence of NAD+ (*t*-test): $p = 0.0265$ at 10 μmol m$^{-2}$ s$^{-1}$ and $p = 0.0033$ at 40 μmol m$^{-2}$ s$^{-1}$. n.s. means statistically no significance ($p = 0.607$ at dark condition). **e** Gross NADP changes relative to NADP pool size in the absence or presence of NAD+ in the dark, as shown in (**d**). **f** pH-dependence of NADP synthesis in ruptured chloroplasts. Data represent mean ± SD ($n = 4$ biological replicates). More details can be found in Supplementary Fig. 2.

activity of *pgr5* under light condition (Supplementary Fig. 7b, $p = 0.541$ *t*-test). Thus, other factors besides NADK activation are involved in de novo NADP+ supply, which is essential for increasing NADP pool size.

## Dark conditions decrease the NADP pool in chloroplasts by activating NADP+ dephosphorylation

In *A. thaliana*, ecotype Col-0 grown at 70 μmol m$^{-2}$ s$^{-1}$, the size of the cellular NADP pool decreased by approximately 50% in nearly 60 min after shading and reached basal levels within 2 h (Supplementary Fig. 8). The time at which the light-regulated amount halved ($t_{1/2}$) was approximately 31.1 and 22.4 min for NADP+ and NADP pool, respectively. As a consequence of the opposite response to light and dark conditions, the NADP pool in chloroplasts fluctuated under a 30 min light/dark cycle (Supplementary Fig. 9). Notably, the dark-induced decrease in NADP pool was initiated after a short delay and the duration varied depending on the light intensity before shading (Fig. 3a; $t_{1/2} = 19.5$ and 10.3 min for plants grown at 70 μmol m$^{-2}$ s$^{-1}$ and 40 μmol m$^{-2}$ s$^{-1}$, respectively). A decrease in NADP pool size was further delayed in proportion to the duration of high-light irradiation (150 μmol m$^{-2}$ s$^{-1}$)

before shading (Fig. 3b, $p < 0.01$, Tukey's HSD test). Similar to shading, DCMU treatment immediately decreased the NADP pool in a dose-dependent manner in plants grown at 70 μmol m$^{-2}$ s$^{-1}$ (Fig. 3c, $p < 0.001$ two-way ANOVA, Supplementary Fig. 10). Furthermore, we found that NAD levels increased with a decrease in NADP pool size (Fig. 3d, $p < 0.01$ two-way ANOVA, Supplementary Fig. 10), indicating that chloroplast NADP pool size is bi-directionally regulated by phosphorylation and dephosphorylation of NAD+ in response to light and dark conditions. We detected NADP+ dephosphorylation (NADPP) activity in the ruptured chloroplasts (Fig. 3e), and unlike NAD+ phosphorylation activity (i.e., NADK activity), NADPP activity was the highest under acidic conditions, whereas activity was not observed under alkaline conditions (Fig. 3f, g). NADPP released phosphate from NADP+ and was inhibited by sodium fluoride (NaF; phosphatase inhibitor) (Supplementary Fig. 11). Moreover, NaF treatment accelerated the light-driven increase in NADP pool size in wildtype plants and recovered from its delay in *pgr5* (Supplementary Fig. 12, $p < 0.05$ *t*-test), suggesting that the dynamics of NADP pool size in response to light is determined by the balance between NAD+ phosphorylation and NADP+ dephosphorylation.

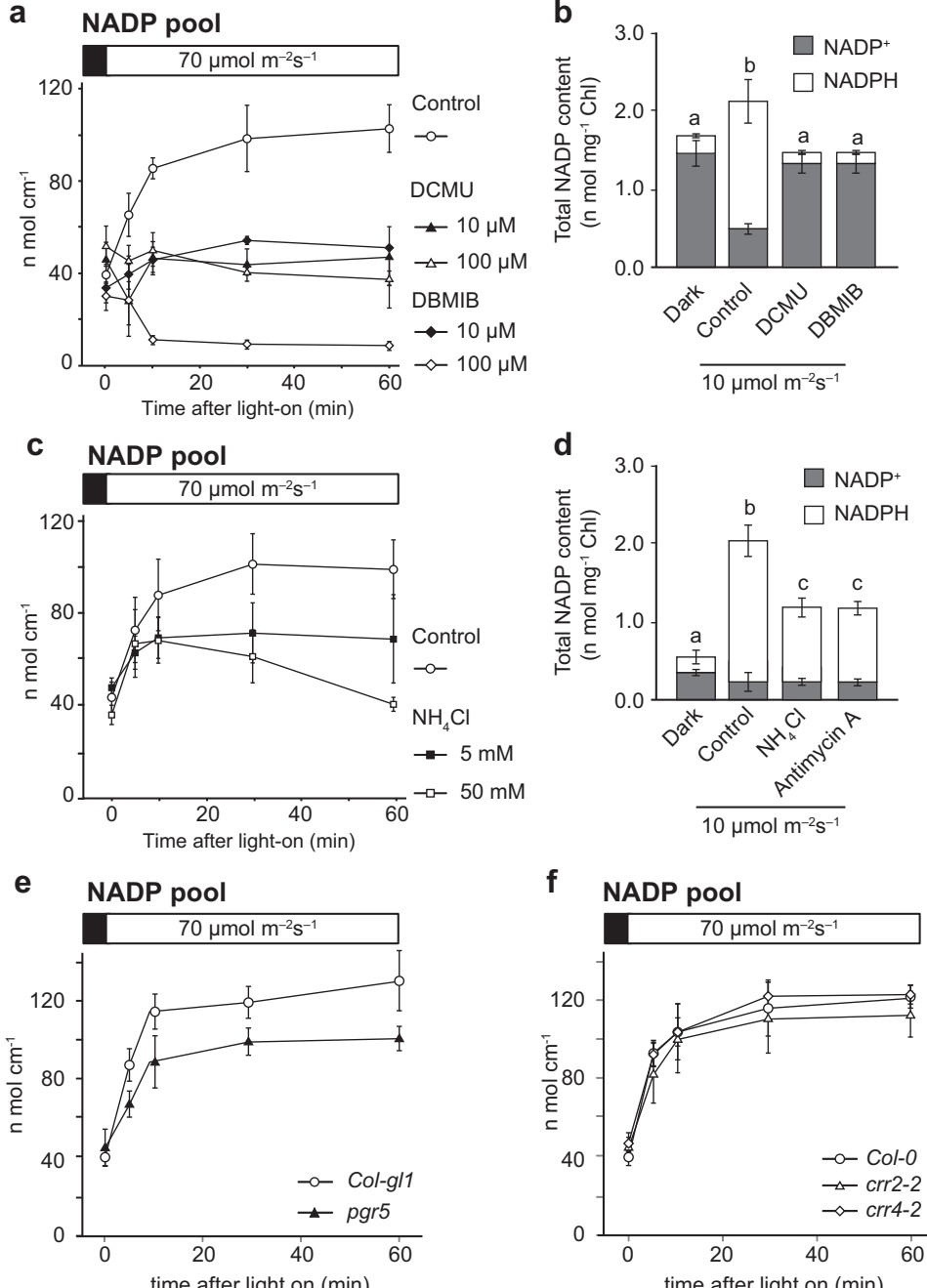

**Fig. 2 | Light-dependent increases in NADP pool size controlled by photochemistry via PGR5/PGRL1-dependent CET. a** Response of NADP pool size to light in the presence of a photochemical inhibitor, DCMU or DBMIB, in leaf discs. Data represent mean ± SD ($n = 3$ biological replicates). **b** Effect of 10 µM DCMU or DBMIB on NADP pool size in isolated chloroplasts 15 min after illumination. There are statistically significant differences between the different letters ($p < 0.05$, Tukey's HSD test). Data represent mean ± SD ($n = 6$ biological replicates). **c** Response of NADP pool size to light in the presence of ammonium chloride (NH₄Cl) in leaf discs. Data represent mean ± SD ($n = 6$ biological replicates). **d** Effect of 5 mM NH₄Cl or 100 µM Antimycin A on NADP pool size in isolated chloroplasts 15 min after illumination. There are statistically significant differences between the different letters ($p < 0.05$, Tukey's HSD test). Data represent mean ± SD ($n = 6$ biological replicates). **e**, **f** Light-dependent responses of NADP pools in CET mutants: PGR5/PGRL1 pathway (**e**) and NDH pathway (**f**). Data represent mean ± SD ($n = 3$ biological replicates). The black bar at the top of the graph indicates dark conditions and the white bar indicates light conditions.

## CET-dependent proton motive force (pmf) associates with the maintenance of NADP pool size in chloroplasts

To confirm the in vivo pH dependence of NADPP activity, we investigated NADP dynamics under dark conditions using CET mutants. *Pgr5* demonstrated remarkably faster kinetics related to decreases in NADP pool size ($t_{1/2} = 4.5$ min), whereas *crr2-2* and *crr4-3* mutants showed comparable kinetics to wildtype (Fig. 4a and Supplementary Fig. 13; $t_{1/2} = 11.0–16.8$ min). In contrast, no clear difference was

observed in the half-life of NADPH between the wildtype and mutants (Supplementary Fig. 14). Thereafter, we investigated mutants with a greater electron flow from stroma that demonstrated proton reflux to Cyt $b_6f$ complex via CET activity. M-type thioredoxins (Trx-m) regulate the PGR5/PGRL1-dependent pathway by directly binding to PGRL1 in a redox-dependent manner[36], and its mutants, *trx-m-124-1* and *trx-m-124-2*, had a slow P700 oxidation rate induced by far-red (FR) light (Fig. 4b), indicating larger electron flow from stroma via

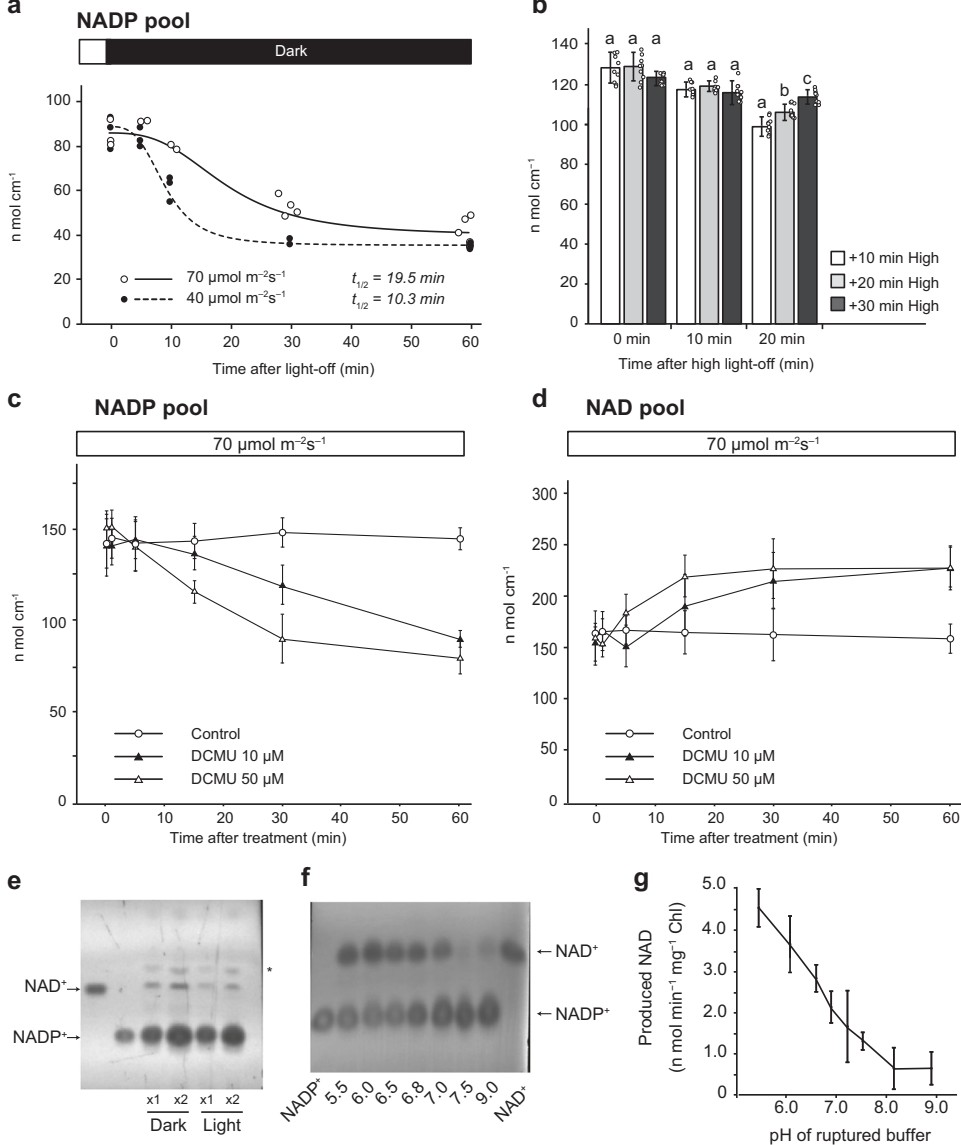

**Fig. 3 | Dark-induced conversion of NADP⁺ into NAD⁺ by NADPP in chloroplasts.**
**a** Time-dependent behavior of the NADP pool size after exposure to dark conditions. Leaf discs were acclimated to 40 μmol m⁻² s⁻¹ (closed circle) or 70 μmol m⁻² s⁻¹ (open circle) light for 2 h before dark exposure. **b** Effect of high-light supplementation at 150 μmol m⁻² s⁻¹ before dark exposure. There are statistically significant differences between the different letters ($p < 0.05$, Tukey's HSD test). Data represent mean ± SD ($n = 9$ biological replicates). **c**, **d** Dose-dependent responses of NADP (**c**) and NAD (**d**) pool sizes to DCMU, a photochemical inhibitor, under continuous light conditions. In this experiment, leaf discs were light-acclimated on a control solution containing 1% ethanol. The solution was replaced with a solution containing DCMU so that light conditions would not change during the procedure and was immediately used for the experiment without infiltration. Data represent mean ± SD ($n = 6$ biological replicates). **e** Detection of NADP⁺ dephosphorylating activity by NADP⁺ feeding to dark- or light-acclimated chloroplasts. Relative loading amounts are indicated below the image. Representative of three independent experiments with similar results was shown. **f** The pH-dependence of NADP⁺ dephosphorylation. Representative of four independent experiments with similar results was shown. **g** Luminescence-based evaluation of the pH-dependence of NADP⁺ dephosphorylation. Data represent mean ± SD ($n = 4$ biological replicates). Arrows in (**e**) and (**f**) refer to NAD⁺ and NADP⁺, respectively, and the asterisk in (**e**) designates an unidentified metabolite detected on the TLC plate.

CET activity. Additionally, the *inap1* mutant, with the inability to transfer electrons to Trx-m protein[17], also showed a reduced oxidation rate of P700 (Fig. 4b). Therefore, these results support that *trx-m-124* mutants are deficient in the downregulation of PGR5-dependent CET activity and have higher CET activity than that of the wildtype, which is consistent with previously reported results[36]. Interestingly, a light-dependent increase in NADP pool size is significantly accelerated in these mutants[17]. In line with this observation, the light-dependent increase in NADP pool size is delayed in *pgr5* and in the presence of Antimycin A (see previous section). Therefore, PGR5-dependent CET activity is involved in the adjustment of NADP pool size in response to light conditions. The rate of decrease in the

NADP pool was decelerated in *trx-m124-1* ($t_{1/2} = 17.2$ min), *trx-m124-2* ($t_{1/2} = 19.4$ min), and *inap1* ($t_{1/2} = 28.3$ min) without any substantial difference in NADPH dynamics between the wildtype and mutants (Fig. 4c and Supplementary Figs. 13 and 14). Although the ΔpH on growth light condition (86 or 167 μmol m⁻² s⁻¹) in the *trx-m 124-1* and *trx-m 124-2* mutants was not greatly enhanced, we observed a considerably higher *pmf* maintained by electrons undergoing reflux from the stromal electron pool to the CET relative to that in Col-0 plants, especially under low-light intensity (31 μmol m⁻² s⁻¹) (Fig. 4d, $p < 0.05$ Tukey's HSD test, Supplementary Fig. 15). This trend was more pronounced in *inap1* mutants and less pronounced in *pgr5* mutants ($p < 0.05$, Tukey's HSD test). Thus, the size of the NADP pool after

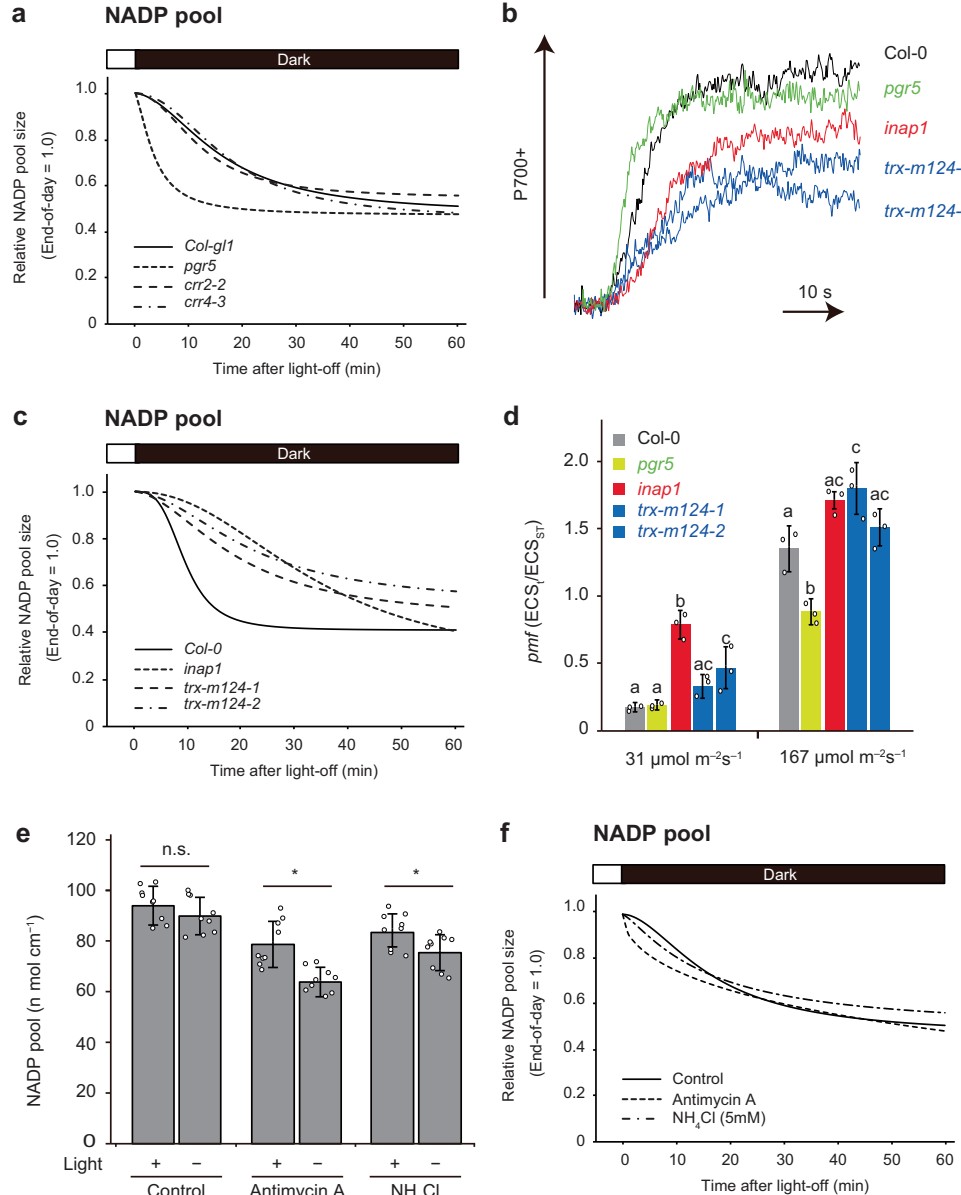

**Fig. 4 | Dark-induced decrease in NADP pool size competes with PGR5/PGRL1-dependent CET. a** Effect of CET deficiency on dark-induced decrease in NADP pool size of leaf discs. Regression curves of relative NADP pool size in leaf discs were shown. More details can be found in Supplementary Figs. 13 and 14. **b** P700 oxidation kinetics of detached leaves during FR exposure. **c** Effect of CET acceleration on dark-induced decrease in NADP pool size of leaf discs. Regression curves of relative NADP pool size in leaf discs were shown. More details can be found in Supplementary Figs. 13 and 14. **d** Total size of the *pmf* of detached leaves. There are statistically significant differences between the different letters ($p < 0.05$, Tukey's HSD test). Data represent mean ± SD ($n = 3$ biological replicates). More details can be found in Supplementary Fig. 15. **e** Changes in NADP pool size of leaf discs in response to 5 min under dark conditions in the presence of antimycin A and ammonium chloride, respectively. Asterisks indicate statistically significant decrease for 5 min darkness (*t*-test): $p = 0.00055$ in antimycin A and $p = 0.0249$ in ammonium chloride. n.s. means statistically no significance ($p = 0.1738$ in Control). Data represent the mean ± SD ($n = 9$ biological replicates). **f** Effect of *pmf* inhibition on dark-induced decreases in the NADP pool size of leaf discs. Regression curves of relative NADP pool size in leaf discs were shown. More details can be found in Supplementary Fig. 16. The black bar at the top of the graph indicates dark conditions and the white bar indicates light conditions.

shading could be temporarily maintained by stromal pH built up via stromal electron pool-derived CET-dependent *pmf*. Consistently, Antimycin A treatment inhibited the maintenance of NADP pool size, even 5 min after shading (Fig. 4e, $p < 0.05$ *t*-test, Fig. 4f and Supplementary Fig. 16). Two envelope transporters, KEA1 and KEA2, adjust stromal pH during light to dark transition[37]. Importantly, the rate of neutralization of stromal pH is significantly impaired in *kea1kea2* mutants after dark transition[37]. Similar to the behavior of stromal pH, the rate of NADP decrease was clearly decelerated in the *kea1-1kea2-1* ($t_{1/2} = 22.4$ min) and *kea1-2kea2-2* ($t_{1/2} = 20.2$ min) mutants (Supplementary Figs. 13 and 14).

## Discussion

Chloroplasts in higher plants rapidly switch between linear and cyclic electron flow in response to changing environmental conditions[38]. This study showed a possible mechanism that this switching is mediated through the quantitative regulation of NADP during photosynthetic induction. We investigated NADP dynamics under dark and light/dark conditions and in response to CET modulation to better understand the mechanism of LET regulation. The results suggested a novel role of ΔpH formation via CET in inducing NADP⁺ synthesis during the photosynthetic induction phase and in the maintenance of NADP pool size during a short period of darkness (i.e., transient shading). De novo

NADP+ production requires (1) NADK2 enzymatic activity, (2) the substrates, NAD+ and ATP, and (3) suitable reaction conditions. In this study, we propose a system in which CET is involved in the regulation of light-responsive interconversion between NADP+ and NAD+ in chloroplasts by adjusting stromal pH conditions.

CET allows electrons to be recycled back from PSI to PQ and forms ΔpH across thylakoid membranes when NADP+ is in short supply, owing to a decrease in NADPH demand under various conditions, such as at low temperatures when the enzymatic activity of the Calvin cycle declines. Thus, CET generates ATP without net NADPH generation and modulates the ATP/NADPH ratio to optimize downstream metabolism[16,39,40]. In addition to ATP supply, the rapid generation of ΔpH controls the induction of non-photochemical quenching and reoxidation of PSI complexes, thereby providing protection against PSII damage caused by excess light and preventing over-reduction of PSI by slowing the turnover by Cyt $b_6f$ complex[41]. Moreover, the PGR5/PGRL1 pathway regulates the LET rate, thereby controlling the flux of electrons and preventing photodamage to PSI by CET under fluctuating light conditions[42]. These results clarify that CET is not just an alternative means of electron flow but also a physiologically important pathway. The present study suggests a potential role of CET in regulating de novo NADP+ supply as the electron acceptor of LET. Luminescence-based NADP quantification revealed NADP+ dynamics in response to light conditions and photochemical inhibitors and enabled the detection of these dynamics in CET mutants.

Consistent with previous studies[17], NADK2 is initially activated by light and redox regulation. However, NADP+ is not fully produced under acidic to neutral pH conditions even after NADK2 activation. In fact, the in vitro NADP-producing activity was >10-fold higher than that observed in organello, suggesting that the activation state of NADK2 is not the only regulator of chloroplastic NADP pool size (see "Results" section). Moreover, in vitro NADK activity was not suppressed in the pgr5 mutants, however, the light-driven increase in NADP pool size was significantly delayed and the steady-state NADP pool size was reduced. Importantly, inhibition of NADP phosphatase partially restored the delay of light-driven increase in NADP pool size. Thus, a rate of NADP pool size increase was adjusted through the balance between NAD+ phosphorylating activity and NADP+ dephosphorylating activity at the beginning of illumination. In addition to the balance control, supply control of ATP could also be involved in the regulation of NADP dynamics because mature chloroplasts do not import cytosolic ATP[43] and the inhibition of NADP phosphatase does not fully restore NADP dynamics. Thus, the PGR5/PGRL1 pathway might contribute to determining the NADP pool size via the ΔpH formation and ATP supply. Further studies are essential for elucidating the contribution of CET to ATP supply in NADP+ synthesis in determining NADP pool size at the steady-state level.

The pH preference for the interconversion between NADP+ and NAD+ was consistent with stromal pH changes: neutral to alkaline under light conditions and slightly acidic in complete darkness (see "Results" section). Delay in light-driven increases in the NADP pool in the pgr5 mutants suggested that the PGR5/PGRL1 pathway (and not the NDH pathway) triggers ΔpH formation by adjusting the stromal environment for NAD+ to NADP+ conversion. Conversely, the dark-induced decrease in NADP pool size was accelerated in the pgr5 mutants while that was decelerated in the mutants, trx-m124 and inap1, with high PGR5-dependent CET activity and pmf, indicating that higher CET activity might increase NADP pool size and less likely to decrease NADP pool size. Depending on the CET activity, electrons temporarily stored in the stroma are refluxed and transferred to P700 for a short duration after shading[44]. These results suggested that ΔpH and stromal pH could be maintained for several minutes, which may also temporarily maintain the NADP pool size. This hypothesis was supported by the fact that kea1kea2 mutants, exhibiting delayed stromal pH neutralization in the dark[30], also show a significant delay of decrease in

NADP pool size. Therefore, photochemistry-regulated stromal pH conditions probably adjust NADP pool size by balancing NAD+ phosphorylation and NADP+ dephosphorylation. A hypothetical model shown in Fig. 5 explains the possible mechanism of NADP homeostasis during photosynthesis under fluctuating natural light conditions, considering that: (1) CET is driven; (2) stromal pH is adjusted suitable for NAD+ phosphorylation and unsuitable for NADP+ dephosphorylation, increasing NADP pool size; (3) LET is fully driven; (4) stromal pH is maintained by reflux of stromal electron pool via CET under transient shading, and maintenance of stromal pH prevents NADP+ dephosphorylation; and (5) NADP pool size is unchanged, even during transient daylight hiding. In contrast, when the stromal electron pool is depleted and ΔpH formation ceases due to prolonged darkness by overcast or dusk conditions, NADP+ dephosphorylation becomes dominant, decreasing the NADP pool size and increasing the NAD pool size. Briefly, the presence of electron reflux via CET may allow plants to recognize the difference between shadow and night conditions. Besides the electron reflux, processes that consume reducing power in the stroma in the dark and ion flux activity are also involved in regulating NADP pool size[45,46], and there will be an integrated control system that includes these processes.

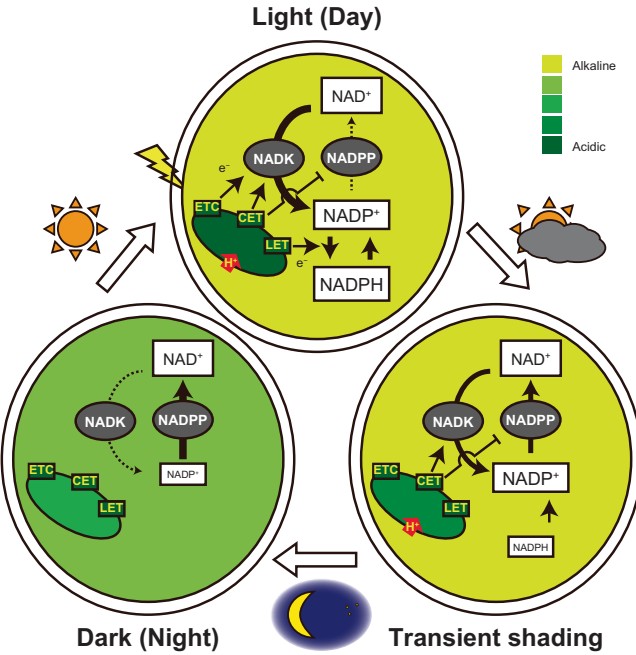

**Fig. 5 | Hypothetical model of NADP pool size regulation via stromal pH condition.** Under dark conditions, NADK (phosphorylates NAD+), is inactive; additionally, the stromal pH is also unfavorable for NADK activity. Furthermore, the stromal pH is relatively favorable for NADPP activity (dephosphorylates NADP+). Consequently, chloroplast NADP pool size is at basal levels in dark conditions. Upon illumination, CET is initially driven to adjust stromal pH, which is favorable for NADK activity (arrows from CET to NADK) and unfavorable for NADPP activity (T-bars from CET to NADPP), by proton (H+) pumping and NADK is further activated by photochemical electron transfer chain (ETC)-dependent redox modification (arrow from ETC to NADK). These modifications promote NAD+ phosphorylation to NADP+, increasing the NADP pool size and preferential driving of LET (arrow from LET to NADK is omitted due to visibility). The LET and CET continue to drive under light conditions for a while, allowing a sufficient amount of electron pool to be stored in the stroma (arrow from LET to NADP redox conversion). Under transient shading, LET halts and NADPH is no longer generated through photochemistry owing to a lack of electron transfer from PSII, increasing NADP+ level. The reflux of the electron pool in the stroma continues to drive CET for several minutes, keeping the stromal pH unfavorable to NADPP activity. When the stromal electron pool is depleted, CET also ceases and the stromal pH becomes favorable for NADPP activity and NADP+ is converted to NAD+, gradually decreasing the NADP pool size.

The discovery of pool size dynamics in this study raises a new question regarding the implications of the decrease in NADP pool size and increase in NAD pool size in the dark. In general, NADP mediates anabolic reactions in the pentose phosphate pathway to support plant growth under dark while NAD is primarily used to produce energy by catabolism[20,22]. The presence of excess $NADP^+$ in the dark might cause undesired glucose consumption in the pentose phosphate pathway, resulting in a shortage of glucose that would normally be supplied to the NAD-dependent TCA cycle and respiratory energy production via glycolysis[47]. The increase in NAD pool size in the dark presumably activates NAD-dependent energy-metabolism pathways. Together with the fact that disturbance of the metabolic regulation of NAD and NADP results in a large change in cellular primary metabolism[48], the NAD and NADP pools are supposed to be regulated to optimize the respective energy metabolism balance between day and night.

In conclusion, we raised a novel role for ΔpH formation via CET in regulating the supply of the electron acceptor $NADP^+$ by adjusting stromal pH. This constitutes a slightly different perspective from the plausible modulation of the ATP/NADPH ratio by ATP supply[49]. The present study offers insight into the quantitative regulation of de novo $NADP^+$ supply in the field of photosynthetic control. Future studies should focus on elucidating the mechanism underlying $NAD^+$ (substrate for $NADP^+$ synthesis) supply to chloroplasts, activation of chloroplastic $NAD^+$ kinase, and the processes associated with $NADP^+$ dephosphorylation.

## Methods

### Plant material and growth conditions

We used *A. thaliana* for all the experiments based on the availability of relevant mutants. During the experiments, the plants were grown in a jiffy pot (Sakata Seed Corporation, Yokohama, Japan) spread on a cultivation tray. Col-0 is a representative wildtype line primarily used in most experiments, and the *crr2-2, crr4-2, inap1, trx-m124-1, trx-m124-2, kea1-1kea2-1,* and *kea1-2kea2-2* mutants were derived from Col-0. Columbia *gl1* (Col-*gl1*) was used as the wildtype line for comparison with the *pgr5* mutant. *pgr5, crr2-2,* and *crr4-2* were used as CET-deficient mutants, and *inap1, trx-m124-1,* and *trx-m124-2* were used as CET-progression mutants. *kea1-1kea2-1* (CS72318) and *kea1-2kea2-2* (CS72319), obtained from the Arabidopsis Biological Resource Center (https://abrc.osu.edu), were used as mutants with high stromal pH even after dark. Plants were grown under a 16 h photoperiod of light intensities of 40 μmol m$^{-2}$ s$^{-1}$ (low light condition) and 70 μmol m$^{-2}$ s$^{-1}$ (usual condition) of fluorescent lamps (Plant Lux; TOSHIBA, Tokyo, Japan) and under 40% to 60% relative humidity and 23 °C air temperature. Arrays of lamp-type white LEDs (NSPW310DS-b2W; Nichia Corp., Tokushima, Japan) were used for high light illumination of 150 μmol m$^{-2}$ s$^{-1}$.

### Luminescence-based quantification of nicotinamide nucleotides

NAD$^+$/NADH and NADP$^+$/NADPH were quantified using the NAD/NADH- and NADP/NADPH-Glo assays (Promega, Chilworth, UK)[31], respectively. Briefly, two leaf discs were prepared from single leaves of 3- to 4-week-old plants for separate extraction of oxidized (NAD$^+$ and NADP$^+$) and reduced forms (NADH and NADPH) and stored in distilled water until further use. Before the experiments, the leaf discs were dark-adapted for 2 h, which appeared to be sufficient to observe a decrease in the NADP pool size to the basal level. Leaf discs were then irradiated with light intensities of 10, 40, or 70 μmol m$^{-2}$ s$^{-1}$. To examine the temperature response, leaf discs irradiated at 70 μmol m$^{-2}$ s$^{-1}$ for 60 min were placed on ice, and irradiation was continued at 70 μmol m$^{-2}$ s$^{-1}$. Sample leaf discs in 0.2 N HCl (for extraction of the oxidized form) or 0.2 N NaOH (for extraction of the reduced form) were immediately boiled for 2 min. After thorough grinding, the extracts were neutralized with sodium phosphate buffer and 0.2 N NaOH or HCl to a pH of approximately 6.0.

### Chloroplast isolation

Leaves of 3- to 4-week-old plants were homogenized in sterile Clp buffer [0.3 M sorbitol, 50 mM HEPES/KOH (pH 7.5), 5 mM EDTA, 5 mM EGTA, 1 mM MgCl$_2$, and 10 mM NaHCO$_3$] passed through a 0.22 μm filter[50], followed by addition of 0.5 mM DTT, mixing with a blender (Nihonseiki kaisha; http://www.nissei-ss.co.jp), and filtration through a sterile cell strainer with 100- and 20-μm nylon mesh (Funakoshi, Tokyo, Japan). After centrifugation for 5 min at 3000×*g*, the pellet was resuspended in Clp buffer and carefully loaded onto 50% Percoll (Sigma–Aldrich, St. Luis, MO, USA) layers for chloroplast isolation.

### Assays for NAD$^+$ phosphorylation and NADP$^+$ dephosphorylation

To measure the in organello NAD$^+$ phosphorylation activity of chloroplasts, isolated intact chloroplasts were resuspended in 50 mM HEPES/KOH and 10 mM MgCl$_2$ containing 0.3 M sorbitol. Three to six chloroplast suspensions acquired in independent isolations were divided into six aliquots for experimental use under three different light conditions with and without the NAD$^+$ precursor. After dark acclimation for 60 min in the presence or absence of 5 mM NAD$^+$, suspensions were illuminated at 10 μmol m$^{-2}$ s$^{-1}$ for 5 min or 15 min or 40 μmol m$^{-2}$ s$^{-1}$ for 15 min and sub-sampled into HCl or NaOH for luminescence-based quantification. To measure the in vitro NAD$^+$ phosphorylation activity of leaf disc, dark-acclimated leaf discs were irradiated for 30 min at 70 μmol m$^{-2}$ s$^{-1}$ in the absence or presence of 10 mM DTT. Leaf discs were ground in liquid nitrogen and dissolved in an extraction buffer containing 50 mM Tricine/KOH (pH 8.8) containing 10 mM MgCl$_2$, and 1 mM phenylmethylsulfonyl fluoride. After centrifuging at 14,000×*g* for 10 min, the supernatants were used for NAD$^+$ phosphorylation activity assay[51]. Briefly, enzyme activity was measured by incubating a reaction mixture containing 50 μg of total protein in the supernatant, 5 mM NAD$^+$, 5 mM ATP, 7 mM MgCl$_2$, and 6 mM nicotinamide at 30 °C for 30 min, followed by quantification of the NADP$^+$ produced. To measure the in vitro NAD$^+$ phosphorylation activity of chloroplasts, isolated intact chloroplasts were suspended in 50 mM HEPES/KOH and 10 mM MgCl$_2$ containing 0.3 M sorbitol. Each of the four chloroplast suspensions was acquired in independent isolations and divided into five aliquots for acclimation to different light intensities (10, 20, 30, 40, or 50 μmol m$^{-2}$ s$^{-1}$). After 30 min of acclimation, the chloroplasts were collected by centrifugation at 2000×*g* for 2 min and ruptured by vigorous resuspension in 50 mM MES/KOH (pH 5.5 or 6.1), 50 mM PIPES/KOH (pH 6.6 or 7.2), 50 mM HEPES/KOH (pH 6.9 or 7.5), or 50 mM Tricine/KOH (pH 8.1 or 8.8) containing 10 mM MgCl$_2$ and 1 mM phenylmethylsulfonyl fluoride, followed by centrifugation at 10,000×*g* for 3 min. The soluble fraction was used for NAD$^+$ phosphorylation activity assay[51].

To detect NADP$^+$ dephosphorylation activity, light-acclimated chloroplasts (at 70 μmol m$^{-2}$ s$^{-1}$) were ruptured, as described, and the soluble fraction was incubated in the presence of 5 mM NADP$^+$ for 30 min in the dark. Detection of NAD$^+$ as a metabolite derived from NADP$^+$ was performed using thin-layer chromatography (TLC) with a solvent system of 5:3 (v/v) isobutyrate and 500 mM NH$_4$OH. Compounds were detected at 254 nm[52], and dephosphorylation activity was assessed by luminescence-based quantification of NAD$^+$. To detect phosphate released from NADP$^+$, protein extracts prepared from dark-acclimated leaf discs in 50 mM MES/KOH (pH 5.5) were incubated with 2.5 mM NADP$^+$ (10 μl) in the presence or absence of NaF (Nacalai Tesque, Inc., Kyoto, Japan) for 15 min at 37 °C. Each sample was then supplemented with 190 μl of extraction buffer and subjected to phosphate quantification assay using malachite green phosphate detection kit (R&D Systems, Inc, Minneapolis, USA).

### Treatment of reagents for leaf discs and isolated chloroplasts

DCMU, DBMIB, ammonium chloride, and Antimycin A were purchased from Sigma–Aldrich and Nigericin was purchased from Nacalai Tesque

Inc. To detect responsiveness to light, leaf discs were vacuum-infiltrated in the dark with a solution containing 10 or 100 μM DCMU, 10 or 100 μM DBMIB, 5 or 50 mM ammonium chloride, 10 mM DTT, 1 mM or 10 mM NaF and 50 μM Nigericin for at least 10 min before illumination. As a control, leaves were treated with 1% (v/v) ethanol (used as a solvent for the inhibitors). For isolated chloroplasts, equal volumes of solution containing 50 mM HEPES/KOH, 10 mM MgCl₂, 0.3 M sorbitol, and 10 mM $NAD^+$ with 20 μM DCMU, 20 μM DBMIB, 10 mM ammonium chloride, or 10, 50, or 100 μM Antimycin A was added to the chloroplast suspension before illumination. Dark-response experiments were initiated immediately after the addition of an equal volume of solution containing 20 or 100 μM DCMU, 10 mM ammonium chloride, or 100 μM Antimycin A to 1% (v/v) ethanol in which the leaf discs were floating under light. Thereafter, leaf discs were incubated under either light or dark conditions.

### In vivo measurement of P700 oxidation kinetics and electrochromic shift (ECS)

For photochemical measurements, three 4-week-old individuals of each line were dark-adapted for 2 h (sufficient to drop the NADP pool size to basal levels), and one leaf from each individual was detached and used in the experiment for reproducibility. P700 oxidation kinetics were monitored using the Dual-PAM-100 system (Walz, Effeltrich, Germany). The redox change of P700 was monitored by measuring the absorbance at 830 nm with relatively high-intensity measuring light (1.3 μmol m⁻² s⁻¹)[53]. FR light was applied to preferentially activate PSI, and the ECS signal was monitored according to a change in absorption at 515 to 550 nm using the Dual-PAM-100 system equipped with a P515/535 emitter-detector module (Walz), as described previously[54]. The ECS signal was obtained after 3 min of illumination at 31, 86, or 167 μmol m⁻² s⁻¹ actinic light (AL), and three 1-s pulses (once every 30 s) were applied to generate technical triplicates. The difference in the full amplitude of the rapid decay of the ECS signal during the dark pulse was determined as $ECS_t$, which represents the difference in total *pmf* between light and dark conditions. Since the change in 515 to 550 nm absorption induced by a single turnover flash ($ECS_{ST}$) on dark-acclimated leaves accounted for variation in leaf thickness and chloroplast density among the leaves, all $ECS_t$ values were normalized against each $ECS_{ST}$ value. The relative partitioning of the *pmf* of ΔpH and ΔΨ was analyzed from the *pmf* parsing traces. The ECS steady state ($ECS_{SS}$) and ECS inverse ($ECS_{inv}$) were extracted from the traces, and ΔpH and ΔΨ were estimated according to a previous report[55].

### Data analysis

Two-sided *t*-test was used to determine the difference due to $NAD^+$ addition in Fig. 1d, the difference of NADP pool size of mutants in Fig. 2e, f, the difference due to light-off in Fig. 4e and the difference of NADK activities in Supplementary Fig. 7b. To determine the significance of the difference between multiple comparison, one-way ANOVA was used in Fig. 1a–c, Supplementary Fig. 7a. To determine the significance of difference of NADP and NAD dynamics between control and chemical treatment, two-way ANOVA was used in Figs. 2a, c and 3c, d. Dunnett's test was used to compare each time point with 0 min while adjusting for multiple comparisons in Fig. 1a, b. Tukey's HSD test was used to determine difference(s) adjusting for multiple comparisons between chemical treatments in Fig. 2b, d, between exposure time of high light intensity in Fig. 3b and between mutants in Fig. 4d. All the statistical tests were performed in XLSTAT software (2019.3.02; Addinsoft, Paris, France). The $t_{1/2}$ was estimated based on a curve fitted using ImageJ2 software (https://imagej.net/).

### Reporting summary

Further information on research design is available in the Nature Portfolio Reporting Summary linked to this article.

## Data availability

The authors declare that the main data supporting the findings of this study are available within the article and its supplementary information files. Source data are provided with this paper.

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

## Acknowledgements

We thank Dr. M. Kono (The University of Tokyo) for their technical support with photochemical detection, Prof. T. Shikanai (Kyoto University) and Dr. Y. Okegawa (Okayama University) for providing *Arabidopsis* mutant seeds, and Dr. T. Jishi (CRIEPI) for setting the light intensities. This work was supported by grants from the Japanese Society for the Promotion of Science [Nos. 20K06695, 19H04715, and 17H05714]. We would like to thank Editage (www.editage.com) for English language editing.

## Author contributions

S.-N.H. conceived, designed and supervised the project. Y.F., C.I. and S.-N.H. performed experiments and analyzed the data. Y.F., C.I. and S.-N.H. wrote the manuscript. M.K.-Y. and S.-N.H. edited the manuscript.

## Competing interests

The authors declare no competing interests.
