## [Peer Review File · Nature Communications]

Adjustment of light-responsive NADP dynamics in chloroplasts by stromal pHReviewer #1 (Remarks to the Author):

In this manuscript, the authors hypothesized that NADP⁺ supply is driven by CET, and tested this hypothesis (line 80). They concluded that CET upregulated NADP⁺ in response to light and downregulated NADP⁺ in response to darkness (Abstract).

I think this hypothesis has not been adequately validated by the data provided in this manuscript. Below are some major issues:

1. A plausible mechanism linking how CET drives NADP⁺ supply must be adequately supported for publication. Stromal NADP⁺ is synthesized by NADK2 using imported NAD⁺ as the substrate. The author did not explain the mechanism on how CET could affect the activities of NADK2 or NAD⁺ import clearly nor provide adequate data to support the claims. For example, while there will be many other changes, besides NADP pool, in the *pgr5* mutant. Could the author conclude that CET drives XXX, without providing the mechanistic link? The author implies that CET leads to alkalization of stroma, which in turns activates NADK. However, LET also leads to alkalization, is it more correct to say both CET and LET drive NADP⁺ synthesis? The authors concluded that "we do not suggest that stromal pH directly dominates NADP pool size in chloroplasts (line 231), given that previous studies demonstrated light- and redox-regulation of NADK2 protein activity"⁷. However, there is no further explanation. Finally, the authors speculated that "photochemistry-regulated stroma pH conditions adjust NADP-synthesis activity under conditions in which light triggers an unidentified mechanism that promotes NADK2 activity and NAD⁺ uptake" (lines 236-238). Hence, the hypothesis is not adequately supported by the data and the mechanism is unclear.

For instance, I could also provide a possible explanation on why the increase in NADP pool is lower in *pgr5* mutant than in WT (Fig. 2e). It has been shown that mature Arabidopsis mesophyll chloroplasts do not import cytosolic ATP (Voon et al., PNAS, 115:E10778, 2018). If the PGR5 pathway does contribute 30% proton gradient across the thylakoid membrane (Ref. 13), the imbalance of stromal ATP/NADPH ratio will cause accumulation of NADPH, a higher NADPH/NADP⁺ ratio, and leads to insufficient NADP⁺ (e.g. Fig. S5). Inadequate NADP⁺ will lead to over-reduction in the stroma and the production of ROS under prolonged illumination (Kozuleva et al., Plant Physiology, 172:1480, 2016 (Fig. 4)). ROS may then inhibit the activities of NADK2, or affect NAD import or synthesis.

2. Redox changes induced by illumination happens within a minute, whereas the changes in NADP pool requires much longer time (> 5 or 10 min). Again, the mechanistic link between a quick response and a slow metabolism changes should be discussed. One important data to support the hypothesis is that AA treatment (Fig. 2d) inhibited the increase in light-driven NADP levels. However, it is not known the time scale of this experiment. As shown in Ref. 13, CET mediated by PGR5 can contribute to delta pH changes in seconds. Ref. 19 (Fig. 4) also shown that when the light was off, the stromal NADPH level in isolated chloroplasts dropped to the basal level in 30 sec. Using an in planta NADPH sensor, stromal NADPH level increased upon illumination, and decreased quickly in 30 sec in darkness (Lim et al. Nature Communications 11:3238, 2020). All these experiments showed that stromal NADPH level responses to illumination quickly. However, many experiments on leaf disks took at least 5 min up to 30, 60 or 120 min (e.g. Fig. 1a-c; Fig. 2a/c/e/f). For those data shown in bar charts (e.g. Fig. 1d, 2b & d, 4e), the duration or experimental time points were not specified. If AA and NH₄Cl indeed inhibited CET and NADP pool, can the authors showed the kinetics of AA and NH₄Cl treatments like Fig. 2c? This would help the reader to evaluate the causal relationship between CET and generation of NADP pool.

3. The authors just grouped many data together to support the hypothesis. More explanation of the following mechanisms should be provided to strengthen the conclusion of this manuscript:

a. Line 134. Please explain the mechanism how ammonium chloride removes Δ pH and membrane potential ($\Delta\psi$).

b. The mechanisms on how *inap1* and *trxm124* mutants affect CET should be explained. Do they interact or regulate NADK2 in a light- or redox-dependent manner?

c. Can you explain why DCMU treatment lower NADP⁺ level but increase NADPH level in Fig. S8?

d. There is no CET in the dark. Can you explain why the NADP level dropped quickly in the *pgr5* mutant than WT in the dark (Fig. S9)?

e. Can you explain how "the size of the NADP pool after shading was temporarily maintained by the pmf and Δ pH" (line 189)? As shown in Fig. S10, the pmf and Δ pH was only maintained for ~30 - 60 sec. How could this affect the half-lives of NADPH in these lines (4.5 - 28 min)?

4. The level of NADP⁺ is a balance of its production by NADK2, its regeneration from NADPH by

CBB/NADP-MDH/fatty acid synthesis (under light) and its conversion to NADPH by LET and the other enzymes (e.g. stromal NADP-malic enzyme). The CBB and NADP-MDH are activated within 30 sec after light is turned on. The author observed that NADPH increased at a faster rate at 40 $\mu\text{mol m}^{-2} \text{s}^{-1}$ than at 10 $\mu\text{mol m}^{-2} \text{s}^{-1}$ (Fig. S1c) but the NADP⁺ kinetics were similar between different light intensities (Fig. S1b) and interpreted that "NADP⁺ level is controlled independent of NADPH" (lines 93-97). While I believe that "NADP⁺ level is controlled independent of NADPH", as NADPH is downstream of NADP⁺, this is not a conclusion drawn from the data of Fig. S1b and S1c. In fact, if more NADP⁺ is converted to NADPH at 40 $\mu\text{mol m}^{-2} \text{s}^{-1}$, more NADP⁺ should have been generated at 40 $\mu\text{mol m}^{-2} \text{s}^{-1}$ (than 10 $\mu\text{mol m}^{-2} \text{s}^{-1}$) too, and a higher portion of that was converted to NADPH, so that the NADP⁺ at equilibria at both light intensities are similar in Fig. S1b. The same consideration should apply to the calculation of NADP-synthesis activity (lines 116-118, Fig. 1e).

5. Many data are presented as NADP pool. Both the data of NADP⁺ and NADPH should be presented (e.g. as supplemental data). For example, Fig. 1c, Fig. 2c, Fig. S9...

Some statements may be misleading or concluded from misinterpretation of data:

1. The authors repeatedly mentioned that at the onset of light exposure, there was insufficient or only a small amount of NADP⁺ (lines 24, 69, 74, 76). However, how low is low? How low is insufficient? In Lines 74-75. The authors wrote "The fast phase indicates the presence of only a small amount of NADP⁺ at the onset of light exposure, whereas the slow phase reflects the limitation of NADP⁺ available for NADPH generation (Ref. 19)". The authors of Ref. 19 did not make the above conclusion. They showed that illumination quickly and significantly increased stromal NADPH level in isolated chloroplasts within 1 sec (fast phase, of which the fluorescence signal increased from 0 to ~0.012 in WT) and in the next 10-20 sec the fluorescence signal increased to ~0.015 (slow phase). I think the slow phase is the time needed for reaching an equilibrium of NADPH production (e.g. LET) and consumption (CBB and light-activated of NADPH-MDH, etc.), rather than due to "the limitation of NADP⁺ available for NADPH generation". Can the authors explain how could this kinetic measurement can lead to a conclusion that "The fast phase indicates the presence of only a small amount of NADP⁺ at the onset of light exposure"? Even if there are a significant amount of NADP⁺ at the onset of light exposure, a rapid increase in NADPH could be seen as well. In Fig. S1, the NADP⁺ level increased from ~33 nmol cm⁻¹ (low?) to ~55 nmol cm⁻¹ in 5 min. Is 33 nmol cm⁻¹ a small amount?

2. Line 107. The authors wrote "Unexpectedly, NADP level decreased by 0.14 nmol mg⁻¹ chlorophyll (Chl) under trace-light (10 $\mu\text{mol m}^{-2} \text{s}^{-1}$) and 0.47 nmol mg⁻¹ Chl under low-light (40 $\mu\text{mol m}^{-2} \text{s}^{-1}$) conditions when intact chloroplasts suspended in isolation buffer were exposed to light for 15 min [Fig. 1d, NAD⁺ (-)]. ". This is not "Unexpectedly". It is reasonable that illumination drove the conversion of NADP to NADPH by FNR, and the conversion is naturally higher at 40 $\mu\text{mol m}^{-2} \text{s}^{-1}$ than at 10 $\mu\text{mol m}^{-2} \text{s}^{-1}$.

3. Lines 203-204. The author wrote "we found that NADP pool size decreased at low temperatures when the Calvin cycle activity declined (Supplementary Fig. 1). Thus, we consider that CET does not produce ATP to generate NADP". What do you mean? What is the causal relationship? Could the slight decrease in NADP pool size at low temperature due to a low activity of NADK2 in low temperature?

Reviewer #2 (Remarks to the Author):

I congratulate the authors to their nice manuscript. I was not aware at all of the possibility that NADP synthesis might be the answer to how plants switch from CET to LET. This finding might make it into the text books on plant physiology. The combination of good methods to quantify NADP pool sizes in combination with using different genotypes affected in CET or NADP synthesis activity provides a convincing picture. I think for some passage the writing could be improved to sell the story even better. I also wonder why NADP levels are so dynamic and ask the authors to share their thoughts what the advantages are of regulating NADP synthesis instead of just keeping levels constantly high. I would suspect that the biosynthetic costs are marginal, so there must be a regulatory advantage. The authors might also rephrase the title of their manuscript to make it catchier.

Details:

P1 Title. In the Discussion section the authors write: "switching is mediated through the quantitative regulation of NADP during photosynthetic induction". Therefore, a much sexier title would be: "The switch between cyclic and linear electron flow is mediated by light-induced NADP synthesis" or similar.

Introduction. Maybe change the order of paragraphs and start first with photosynthesis, CET and end with the unexplored role of NADP synthesis?

P38: „promising“ what do you mean? Possible, plausible, etc.? Please rephrase.

P49: strictly speaking electrons are transported from water to Fd in LET. Maybe this should be rephrased and not be seen from the PSI view only.

P57: it would be fair to give credit to the people who discovered PGR5 and PGRL1. If citing reviews, indicate that this is a review. This holds true throughout the manuscript.

P70: lack of drive: rephrase

P80: "We hypothesized": in this study or in a previous publication. Better make clear.

P152: In Columbia-0 (Col-0): this is an unusual way to phrase this. It is *A. thaliana*, ecotype Col-0.

P168: a larger stromal electron pool: I think I know what is meant but better to rephrase.

P181: explain the different mutants a bit better. Why are they used? Explain their relevance with respect to NADP levels such that it is clear to the reader without searching the introduction or PubMed.

P238-241: these two sentences do in principle very nicely summarize the main finding of the study. Why not placing this at the beginning of the discussion and explaining then the details? Discussion in general: I wonder why there is a regulation of NADP levels at all, meaning why decreasing it during the night such that CET is need to start LET? Why not keeping NADP levels continuously high? Please extend your discussion in this direction.

Response to Reviewer #1

Comment 1:

A plausible mechanism linking how CET drives NADP⁺ supply must be adequately supported for publication. Stromal NADP⁺ is synthesized by NADK2 using imported NAD⁺ as the substrate. The author did not explain the mechanism on how CET could affect the activities of NADK2 or NAD⁺ import clearly nor provide adequate data to support the claims. For example, while there will be many other changes, besides NADP pool, in the pgr5 mutant. Could the author conclude that CET drives XXX, without providing the mechanistic link?

Response 1:

Thank you for your insightful comment. We too, think it is difficult to conclude that CET drives NADP⁺ supply without a mechanistic link. However, our aim was not to make such a claim in the previous version. However, thanks to your suggestion, we realized that we made an error stating, "We hypothesized that the supply of NADP⁺ is driven by CET" in the introduction. As titled in the revised manuscript, "Adjustment of light-responsive NADP dynamics in chloroplasts by stromal pH regulated by cyclic electron transfer", our novel finding is the importance of Δ pH and stromal pH established by CET in NADP pool size adjustment. It has been proposed that 1) activation of NADK, 2) substrate supply, and 3) regulation of suitable reaction conditions are important for NADP⁺ synthesis, and this paper focuses on the 3rd point and its relationship to CET. To clarify this assertion, we have added a model diagram (Fig. 5) and rewritten the main parts of the paper.

Comment 2:

The author implies that CET leads to alkalization of stroma, which in turns activates NADK. However, LET also leads to alkalization, is it more correct to say both CET and LET drive NADP⁺ synthesis? The authors concluded that "we do not suggest that stromal pH directly dominates NADP pool size in chloroplasts (line 231), given that previous studies demonstrated light- and redox-regulation of NADK2 protein activity". However, there is no further explanation.

Response 2:

We agree that both CET and LET lead to stromal alkalization, as said by the reviewer. Our manuscript describes a phenomenon at the onset of photosynthesis, when the organism receives light for the first time after dark acclimation. Joliot & Joliot (2005) reported that LET activity is much lower than CET activity at this time. Since our previous and current results show that the

NADP pool size in the dark is comparable to that of the chloroplastic NADP⁺ synthesis mutants (*nadk2* mutant), and that there is less NADP⁺ (electron acceptor for LET). Thus, it seems reasonable to assume that LET activity is initially low and CET activity is high.

The statement “Since previous studies have demonstrated light and redox regulation of NADK2 protein activity, it is unlikely that stromal pH directly dominates NADP pool size in chloroplasts (line 231)” means that it is not regulated by stromal pH alone, but also by light activation of NADK2. We have deleted this sentence, substantially rewritten the discussion section, and added the hypothetical model in Fig. 5.

Comment 3:

Finally, the authors speculated that “photochemistry-regulated stroma pH conditions adjust NADP-synthesis activity under conditions in which light triggers an unidentified mechanism that promotes NADK2 activity and NAD⁺ uptake” (lines 236-238). Hence, the hypothesis is not adequately supported by the data and the mechanism is unclear.

For instance, I could also provide a possible explanation on why the increase in NADP pool is lower in *pgr5* mutant than in WT (Fig. 2e). It has been shown that mature Arabidopsis mesophyll chloroplasts do not import cytosolic ATP (Voon et al., PNAS, 115:E10778, 2018). If the PGR5 pathway does contribute 30% proton gradient across the thylakoid membrane (Ref. 13), the imbalance of stromal ATP/NADPH ratio will cause accumulation of NADPH, a higher NADPH/NADP⁺ ratio, and leads to insufficient NADP⁺ (e.g. Fig. S5). Inadequate NADP⁺ will lead to over-reduction in the stroma and the production of ROS under prolonged illumination (Kozuleva et al., Plant Physiology, 172:1480, 2016 (Fig. 4)). ROS may then inhibit the activities of NADK2, or affect NAD import or synthesis.

Response 3:

Thank you very much for providing an elaborate explanation regarding the mechanism of NADP shortage in *pgr5* mutants. As described in Response 1, it has been proposed that 1) activation of NADK, 2) substrate supply, and 3) regulation of suitable reaction conditions are important for NADP⁺ synthesis, and this study was concerned with the 3rd point and its relationship with CET. However, we agree that it is important to know more about the mechanisms involved in regulating NADK2 activity and NAD⁺ uptake. We have performed additional experiments and added the results to the revised manuscript. In brief, firstly, *in vitro* NADP⁺ synthesis assays showed that photochemical inhibitor impaired photoactivation of the NADK (Supplementary Fig. 6a). Secondly, NADK photoactivation is not inhibited in *pgr5* mutants (Supplementary Fig. 6b). Lastly, in spite of *in vitro* NADK activity, *pgr5* had lower NADP⁺ synthesis activity in chloroplasts and antimycin A inhibited NADP⁺ synthesis in

chloroplasts of wildtype (Supplementary Fig. 5). These results strongly suggest that NADK is fully activated even in the CET mutants; however, NADP⁺ synthesis is regulated by reactions downstream of CET. These results narrowed down the reasons for impaired NADP⁺ synthesis in *pgr5* to either substrate supply or reaction conditions.

As the reviewer explains, we also believe that the main reason for the decrease in NADP pool size during the stable phase may be ATP deficiency. Alternatively, the uptake of NAD⁺ into chloroplasts might be reduced in the *pgr5* mutant. To clarify these possibilities, we attempted to construct an experimental system to determine whether NADP synthesis is restored by adding small amounts of NAD⁺ or ATP to isolated *pgr5* chloroplasts, raptured in alkaline solution after NADK2 activation by illumination. Unfortunately, we cannot completely avoid light changes during sample preparation and centrifugation steps; thus, it was not possible to determine the effect of small amounts of ATP and NAD⁺ addition by the current method. In addition, even when NADP measurement was performed directly without sample purification to avoid light changes, the results of NADP measurement highly varied due to debris carryover. As a result, we proposed that substrate supply is involved in the regulation of NADP synthesis.

In the previous manuscript, we showed that NADP⁺ is dephosphorylated to NAD⁺ under acidic to neutral conditions. In this revision, we performed an additional experiment and showed that phosphatase inhibitor NaF promoted light-induced NADP pool size increase in the wildtype and partially restored the delay of NADP pool size increase in *pgr5*. Thus, in this paper, we proposed a hypothesis that in *pgr5*, the regulation of the balance between NAD⁺ phosphorylation and NADP⁺ dephosphorylation, rather than ATP or NAD⁺ shortages, was disturbed at early response in illumination. The balance of NAD⁺ and NADP⁺ interconversion would be regulated by stromal pH downstream of CET, which is reflected in the NADP pool size. To further confirm this, we conducted another experiment using *kea1kea2* double mutants, showing delayed neutralization of stroma pH under dark. The results revealed a delay in the decrease of NADP pool size.

Comment 4:

Redox changes induced by illumination happens within a minute, whereas the changes in NADP pool requires much longer time (> 5 or 10 min). Again, the mechanistic link between a quick response and a slow metabolism changes should be discussed. One important data to support the hypothesis is that AA treatment (Fig. 2d) inhibited the increase in light-driven NADP levels. However, it is not known the time scale of this experiment. As shown in Ref. 13, CET mediated by PGR5 can contribute to delta pH changes in seconds. Ref. 19 (Fig. 4) also shown that when the light was off, the stromal NADPH level in isolated chloroplasts dropped to the basal level in 30 sec. Using an in planta NADPH sensor, stromal NADPH level increased upon illumination,

and decreased quickly in 30 sec in darkness (Lim et al. Nature Communications 11:3238, 2020). All these experiments showed that stromal NADPH level responses to illumination quickly. However, many experiments on leaf disks took at least 5 min up to 30, 60 or 120 min (e.g. Fig. 1a-c; Fig. 2a/c/e/f). For those data shown in bar charts (e.g. Fig. 1d, 2b &d, 4e), the duration or experimental time points were not specified. If AA and NH₄Cl indeed inhibited CET and NADP pool, can the authors showed the kinetics of AA and NH₄Cl treatments like Fig. 2c? This would help the reader to evaluate the causal relationship between CET and generation of NADP pool.

Response 4:

Thank you for your helpful suggestion. Firstly, we mentioned that the NADP measurement of chloroplasts was performed 15 min after irradiation in the methods section; moreover, we have added this in the figure legends in the revised manuscript. Secondly, we do not claim that changes in the NADP pool require longer durations (more than 5 or 10 min). We only suggested that it takes about 30 min to increase this to a steady state. In fact, the increase is observed 1 minute after light exposure (Supplementary Fig. 11). Thus, both redox and metabolic changes occur within at least 1 minute of illumination. Due to the limitation of the experimental technique, sampling quicker than this is challenging. Therefore, we do not believe that the results are significantly inconsistent with the rate of redox change. We also believe that there is a small amount of NADP pool even in the dark and that NADPH originating from this NADP pool is quickly detected by the NADPH sensor. However, based on our experimental data, the actual amount of NADPH may be larger than that measured by the NADPH sensor. Further investigation regarding this will be covered in subsequent studies.

Lastly, we showed the kinetics of NH₄Cl treatment in Fig. 2c of the previous version. However, we have not performed experiments like Fig. 2c with AA. Because AA affects the mitochondrial respiratory chain, it is impossible to distinguish between the effects of mitochondrial respiratory chain inhibition from those of CET inhibition in experimental systems that require pre-treatment before irradiation. Instead, the effects of AA on isolated chloroplasts were examined under 5- and 15-min light irradiations with new results (Supplementary Fig. 6). Similar to the results already shown, it is clear that the NADP pool increase is inhibited by AA, and furthermore, the NADP pool increase is also inhibited in *pgr5* isolated chloroplasts.

Comment 5:

The authors just grouped many data together to support the hypothesis. More explanation of the following mechanisms should be provided to strengthen the conclusion of this manuscript.

a. Line 134. Please explain the mechanism how ammonium chloride removes ΔpH and membrane potential ($\Delta\psi$).

Response 5:

Thank you for your suggestion regarding the role of ammonium chloride in establishing ΔpH and membrane potential ($\Delta\psi$). According to Crofts (1967) and Opanasenko et al. (2010), NH_4 inhibited the light-induced H^+ uptake. We have revised the statement to "We observed a similar inhibitory effect on NADP pool size with the use of ammonium chloride, which inhibits proton pump activity during photochemical electron flow" and added these citations.

Comment 6:

The mechanisms on how *inap1* and *trxm124* mutants affect CET should be explained. Do they interact or regulate NADK2 in a light- or redox-dependent manner?

Response 6:

We added a brief description of how *inap1* and *trxm124* mutants affect CET. According to Okegawa et al. (2020), m-type thioredoxin (Trx-m) is known to directly regulate the PGR5/PGRL1-dependent pathway by binding with PGRL1 in a redox-dependent manner. The *inap1* mutant is a mutant of Fd-thioredoxin reductase and shows defective Trx-m interaction and electron transfer to Trx-m protein.

Comment 7:

Can you explain why DCMU treatment lower NADP^+ level but increase NADPH level in Fig. S8?

Response 7:

Our data (Supplementary Fig. 10) show that the level of NADPH decreases once in response to DCMU and then recovers to the original level. This phenomenon may indicate that the supply of *de novo* NADP^+ is regulated independently of NADPH production. While we agree that this recovery is very interesting, we would like to concentrate on the change in NADP pool size in this paper and leave the discussion of the mechanism of this decrease for subsequent studies; however, we believe that the oxidative pentose phosphate pathway may be involved. As shown in Supplementary Fig. 8, NADPH gradually increases with increasing duration of dark conditions. It is possible that this phenomenon was induced earlier by the low concentration of DCMU treatment.

Comment 8:

There is no CET in the dark. Can you explain why the NADP level dropped quickly in the *pgr5*

mutant than WT in the dark (Fig. S9)?

Response 8:

Thank you for your question. Indeed, the complete CET sequence does not occur in the dark. Shading would stop the electron supply from PSII; however, the supply from the remaining electron pool in the stroma would continue. This is the principle of CET activity, commonly measured in the P700 oxidation assay. It has been repeatedly reported that P700 oxidation, as measured here, is faster in *pgr5*. A smaller cyclic flow from the stroma electron pool (i.e., CET) could explain the rapid Δ pH dissipation and decrease in NADP.

Comment 9:

Can you explain how “the size of the NADP pool after shading was temporarily maintained by the pmf and Δ pH” (line 189)? As shown in Fig. S10, the pmf and Δ pH was only maintained for ~30 - 60 sec. How could this affect the half-lives of NADPH in these lines (4.5 – 28 min)?

Response 9:

Thank you for your question. Firstly, I have added Supplementary Fig. 13 to show the half-lives of NADPH in mutant lines. There was no difference between the wild type and mutants. Thus, the difference in the rate of NADP pool size decrease is due to the difference in the rate of NADP⁺ decrease.

Since experimental conditions, including light intensities, were different between NADP measurement and *pmf* evaluation, it is difficult to directly compare and fully explain the mechanism of temporary maintenance of NADP pool size by pmf and Δ pH. Therefore, in our original manuscript, we described that “the size of the NADP pool after shading was temporarily maintained by the pmf and Δ pH.” However, to avoid confusion, we have revised it to “the size of the NADP pool after shading could be temporarily maintained by the *pmf* and stromal pH”.

One possible explanation is described below and in the discussion section. There must be a variety of signal transductions and regulations after dark. The delay in the decay of pmf and Δ pH could have magnified the delay in downstream processes. In general, it is possible that a delay of a few seconds in the initial reaction could be magnified in a later reaction system: for example, a traffic jam can be caused by the transient braking of the car in front of you. In the case of NADP decrease under dark conditions, we think that the timing of NADK2 inactivation and NADPP activation, or the as yet unidentified regulatory components, resulted in a significant difference. Further studies can focus on unveiling the relationship between pmf/ Δ pH and NADP dynamics. We have acknowledged this and suggested it as a topic for further

research in the discussion section of the revised manuscript. Detailed analysis of enzyme activity using recombinant proteins is required; however, for the NADK2 protein, the function and significance of the N-extension domain are not yet clear, and for NADPP, the associated enzyme has not been isolated. We are currently working on identifying the enzyme genes that contribute to NADP dephosphorylation, in hope that this would help in further explaining the phenomenon.

Comment 10:

The level of NADP⁺ is a balance of its production by NADK2, its regeneration from NADPH by CBB/NADP-MDH/fatty acid synthesis (under light) and its conversion to NADPH by LET and the other enzymes (e.g. stromal NADP-malic enzyme). The CBB and NADP-MDH are activated within 30 sec after light is turned on. The author observed that NADPH increased at a faster rate at at 40 $\mu\text{mol m}^{-2} \text{s}^{-1}$ than at 10 $\mu\text{mol m}^{-2} \text{s}^{-1}$ (Fig. S1c) but the NADP⁺ kinetics were similar between different light intensities (Fig. S1b) and interpreted that “NADP⁺ level is controlled independent of NADPH” (lines 93-97). While I believe that “NADP⁺ level is controlled independent of NADPH”, as NADPH is downstream of NADP⁺, this is not a conclusion drawn from the data of Fig. S1b and S1c. In fact, if more NADP⁺ is converted to NADPH at 40 $\mu\text{mol m}^{-2} \text{s}^{-1}$, more NADP⁺ should have been generated at 40 $\mu\text{mol m}^{-2} \text{s}^{-1}$ (than 10 $\mu\text{mol m}^{-2} \text{s}^{-1}$) too, and a higher portion of that was converted to NADPH, so that the NADP⁺ at equilibria at both light intensities are similar in Fig. S1b. The same consideration should apply to the calculation of NADP-synthesis activity (lines 116-118, Fig. 1e).

Response 10:

We apologize for these errors. Since "NADP⁺ level" and "NADP level" are confusing expressions, we changed them to "*de novo* NADP⁺ supply" and "NADP pool size," respectively. In this manuscript, we focused on the sum of NADP⁺ and NADPH (NADP pool size). The biosynthesis of NADP⁺ from NAD⁺ is a unique pathway to increase the NADP pool size, because NADP⁺ generation from NADPH by CBB/NADP-MDH is unable to increase the NADP pool size. Of course, the change in NADP pool size (Fig. 1d and Fig. 1e) results from the sum of NADP⁺ and NADPH. Therefore, possible interconversions between NADP⁺ and NADPH during experimental procedure was already taken into consideration to calculate NADP-synthesis activity in all of the experiments.

Comment 11:

Many data are presented as NADP pool. Both the data of NADP⁺ and NADPH should be presented (e.g. as supplemental data). For example, Fig. 1c, Fig. 2c, Fig. S9...

Response 11:

As you have suggested, both the data of NADP⁺ and NADPH would be helpful for readers who are interested in redox status rather than the NADP pool size. We have added data on NADP⁺ and NADPH in the supplementary Figs and data resource.

Comment 12:

Some statements may be misleading or concluded from misinterpretation of data:

1. The authors repeatedly mentioned that at the onset of light exposure, there was insufficient or only a small amount of NADP⁺ (lines 24, 69, 74, 76). However, how low is low? How low is insufficient?

Response 12:

Thank you for your insightful comment. Please note that the extent of the low NADP pool in the dark is due to a comparison with the *nadk2* mutant, and has been stated multiple times. We have cited papers showing that the *nadk2*, a mutant of the chloroplast NADP⁺ synthase NADK2, does not increase NADP⁺ and NADPH when exposed to light, that NADK2 enzyme activity is comparable to *nadk2* in the wildtype in the dark, and that the NADP pool size in the wildtype in the dark is comparable to that in *nadk2* mutants. Considering the amount of NADP present outside of chloroplasts (which presently is impossible to measure separately) and the increase in NADP after light exposure, we think that NADP in chloroplasts in the dark is insufficient compared to that under maximum photosynthetic activity.

Comment 13:

In Lines 74-75. The authors wrote “The fast phase indicates the presence of only a small amount of NADP⁺ at the onset of light exposure, whereas the slow phase reflects the limitation of NADP⁺ available for NADPH generation (Ref. 19)”. The authors of Ref. 19 did not make the above conclusion. They showed that illumination quickly and significantly increased stromal NADPH level in isolated chloroplasts within 1 sec (fast phase, of which the fluorescence signal increased from 0 to ~0.012 in WT) and in the next 10-20 sec the fluorescence signal increased to ~0.015 (slow phase). I think the slow phase is the time needed for reaching an equilibrium of NADPH production (e.g. LET) and consumption (CBB and light-activated of NADPH-MDH, etc.), rather than due to “the limitation of NADP⁺ available for NADPH generation”. Can the authors explain how could this kinetic measurement can lead to a conclusion that “The fast phase indicates the presence of only a small amount of NADP⁺ at the onset of light exposure”? Even if there are a significant amount of NADP⁺ at the onset of light exposure, a rapid increase

in NADPH could be seen as well.

Response 13:

Thank you for pointing out that Ref. 19 in the former version may have been an unnecessary citation. We agree that it is difficult to deduce NADP⁺ deficiency from the rapid increase of NADPH fluorescence. We have previously shown that NADP⁺ is synthesized after illumination and that the NADP pool size increases, and this has been confirmed to be reproducible by other groups, i.e. Thormahlen et al., (2017) (Ref6 in former version).

Comment 14:

In Fig. S1, the NADP⁺ level increased from ~33 nmol cm⁻¹ (low?) to ~55 nmol cm⁻¹ in 5 min. Is 33 nmol cm⁻¹ a small amount?

Response 14:

Thank you for pointing this out. sFig. 1b and sFig. 1c had serious errors on the vertical axis, which have now been corrected. Yes, we think ~33 nmol cm⁻¹ is low because the chloroplasts appear to contain 70–80% of the total NADP pool. Even when 55 nmol cm⁻¹ is set at 100%, 11–16.5 nmol cm⁻¹ NADP is outside the chloroplasts, and under dark, out of ~33 nmol cm⁻¹, 11–16.5 nmol cm⁻¹ NADP is outside the chloroplasts. Since the percentage of NADP pool size in chloroplasts is estimated under higher light intensity conditions than our experimental conditions (10 and 40 μmol m⁻² s⁻¹), we would expect the actual NADP pool size in chloroplasts under dark to be much lower. We believe that precise quantification of chloroplast NADP levels and other NADP levels should be addressed in the future.

Comment 15:

Line 107. The authors wrote “Unexpectedly, NADP level decreased by 0.14 nmol mg⁻¹ chlorophyll (Chl) under trace-light (10 μmol m⁻² s⁻¹) and 0.47 nmol mg⁻¹ Chl under low-light (40 μmol m⁻² s⁻¹) conditions when intact chloroplasts suspended in isolation buffer were exposed to light for 15 min [Fig. 1d, NAD⁺ (-)]. “. This is not “Unexpectedly”. It is reasonable that illumination drove the conversion of NADP to NADPH by FNR, and the conversion is naturally higher at 40 μmol m⁻² s⁻¹ than at 10 μmol m⁻² s⁻¹.

Response 15:

We apologize for the confusion caused due to our poor description. As described in **Response 10**, we changed "NADP⁺ level" to "*de novo* NADP⁺ supply" and "NADP level" to "NADP pool size," respectively. As the reviewer mentioned, illumination facilitates the conversion of NADP⁺

to NADPH by FNR. If we say that "unexpectedly" NADP⁺ decreases instead of NADPH increases, it will lead to confusion. Fig. 1d shows that the sum of NADP⁺ and NADPH (the NADP pool) is reduced when NAD⁺ is not added.

Comment 16:

Lines 203-204. The author wrote “we found that NADP pool size decreased at low temperatures when the Calvin cycle activity declined (Supplementary Fig. 1). Thus, we consider that CET does not produce ATP to generate NADP⁺”. What do you mean? What is the causal relationship? Could the slight decrease in NADP pool size at low temperature due to a low activity of NADK2 in low temperature?

Response 16:

Thank you for pointing out that the description was ambiguous. After reconsidering the inevitability within the paper, result and description about low temperature have been removed. One possible role for CET is to provide ATP for NADP⁺ synthesis. For example, it is possible that when NADPH does not convert back to NADP⁺ due to reduced enzymatic activity at low temperatures, there may be a mechanism to safely conserve electrons by using ATP derived from CET to make NADP⁺. In our data, we did not observe an increase but rather a decrease in the NADP pool at low temperatures, thus ruling out this possibility. Unfortunately, it is not possible to discuss at this stage whether the decrease in NADK2 activity at low temperature decreases the NADP pool size. At least some degradation or dephosphorylation of NADP must occur. However, it is unclear how the NADP pool changes if these enzymes, in general, are also less active at low temperatures. Future studies on the dynamics of the NADP pool in response to light and temperature, including identification of pathways and isolation and identification of the enzymes involved, are essential.

Response to Reviewer #2:

Comment 1:

I congratulate the authors to their nice manuscript. I was not aware at all of the possibility that NADP synthesis might be the answer to how plants switch from CET to LET. This finding might make it into the text books on plant physiology. The combination of good methods to quantify NADP pool sizes in combination with using different genotypes affected in CET or NADP synthesis activity provides a convincing picture. I think for some passage the writing could be improved to sell the story even better. I also wonder why NADP levels are so dynamic and ask the authors to share their thoughts what the advantages are of regulating NADP

synthesis instead of just keeping levels constantly high. I would suspect that the biosynthetic costs are marginal, so there must be a regulatory advantage. The authors might also rephrase the title of their manuscript to make it catchier.

Response 1:

I am honored to receive such wonderful comments. We, too, wonder why NADP levels are so dynamic, and we plan to focus on this for our future studies. At present, our hypothesis is that the metabolic switch from anabolism in the light to catabolism in the dark is important. Under light conditions, photochemistry produces ATP and NADPH for anabolic reactions. Under dark, plants have to generate ATP through catabolism. If NADP⁺ remains high under dark conditions, the oxidative pentose phosphate cycle could proceed and consume glucose, instead of providing for glycolysis and mitochondrial respiration. In brief, we consider that regulation of NAD and NADP pool size acts as *en block* switch for a comprehensive metabolic state change. We plan to focus on this in the subsequent studies

Thank you for suggesting to make the title catchy; however, as there is still work to be done, we have rephrased the title of the revised manuscript to “Adjustment of light-responsive NADP dynamics in chloroplasts by stromal pH regulated by cyclic electron transfer”.

Comment 2:

P1 Title. In the Discussion section the authors write: “switching is mediated through the quantitative regulation of NADP during photosynthetic induction”. Therefore, a much sexier title would be: “The switch between cyclic and linear electron flow is mediated by light-induced NADP synthesis” or similar.

Response 2:

Thank you very much for this suggestion. After discussion with all the authors and considering all the comments from reviewers and editor, we re-titled the revised manuscript as “Adjustment of light-responsive NADP dynamics in chloroplasts by stromal pH regulated by cyclic electron transfer”.

Comment 3:

Introduction. Maybe change the order of paragraphs and start first with photosynthesis, CET and end with the unexplored role of NADP synthesis?

Response 3:

Thank you for the advice. We have rearranged the order of the paragraphs as suggested and

believe that this revision has made the subject of our manuscript easier to understand.

Comment 4:

P38: „promising“ what do you mean? Possible, plausible, etc.? Please rephrase.

Response 4:

Thank you for your suggestion. The word "promoting" was ambiguous and has been replaced with "identified" in the revised manuscript.

Comment 5:

P49: strictly speaking electrons are transported from water to Fd in LET. Maybe this should be rephrased and not be seen from the PSI view only.

Response 5:

Thank you for your suggestion. We have accordingly rephrased the description as follows: "When light is absorbed by plant leaves, two photosynthetic protein complexes, photosystem II (PSII) and photosystem I (PSI), are activated; PSII oxidizes water and PSI reduces NADP^+ , via a process called linear electron transfer (LET)."

Comment 6:

P57: it would be fair to give credit to the people who discovered PGR5 and PGRL1. If citing reviews, indicate that this is a review. This holds true throughout the manuscript.

Response 6:

Thank you very much for your guidance. We regret not adding the original citations of the previous studies. This has been corrected in the revised manuscript to an extent.

Comment 7:

P70: lack of drive: rephrase

Response 7:

We rephrased it as follows: "NADPH levels are minimal in chloroplasts in the dark (photosynthesis does not occur), and during this time, NADP^+ also does not accumulate in the chloroplasts; however, it is present at minimal concentrations in chloroplasts¹³, and it is unclear if this minimal NADP^+ is sufficient to drive the LET at the onset of photosynthetic electron transfer."

Comment 8:

P80: “We hypothesized”: in this study or in a previous publication. Better make clear.

Response 8:

Thank you for your advice. This error has been rectified.

Comment 9:

P152: In Columbia-0 (Col-0): this is an unusual way to phrase this. It is *A. thaliana*, ecotype Col-0.

Response 9

Thank you for your suggestion. We have corrected it to “In *A. thaliana*, ecotype Col-0.”

Comment 10:

P168: a larger stromal electron pool: I think I know what is meant but better to rephrase.

Response 10:

We rephrased it to “a greater electron flow from stroma”.

Comment 11:

P181: explain the different mutants a bit better. Why are they used? Explain their relevance with respect to NADP levels such that it is clear to the reader without searching the introduction or PubMed.

Response 11:

I apologize for the inconvenience caused by the lack of explanation. In the revised manuscript, we described the relationship between Trx-m and PFR5/PGRL pathway and the relationship between *inap1* and Trx-m. In addition, we cited and mentioned the accelerated light-responsive increase in NADP pool size in these mutants in a previous report.

Comment 12:

P238-241: these two sentences do in principle very nicely summarize the main finding of the study. Why not placing this at the beginning of the discussion and explaining then the details?

Discussion in general: I wonder why there is a regulation of NADP levels at all, meaning why decreasing it during the night such that CET is need to start LET? Why not keeping NADP

levels continuously high? Please extend your discussion in this direction.

Response 12:

Thank you for your suggestion. We have accordingly changed the structure of the discussion. In addition, we have added a model diagram in Fig. 5 to demonstrate our hypothesis.

Reviewer #1 (Remarks to the Author):

In the revised version, the authors provided additional experimental data (Fig. S5 on AA treatment, Fig. S7 on NAD⁺ phosphorylation, Fig. S11/S12 on NaF treatment, Fig. S13/14 on *kea1kea2* mutants, etc.) and formulated a model (Fig. 5).

In this study, the authors hypothesized that “CET is involved in *de novo* NADP⁺ supply” (line 70) and intended to justify that “CET helps regulate NADP pool size via stromal pH regulation under fluctuating light conditions” (Abstract). As shown in Fig. S9, the amount of NADP⁺ increased under light and decreased under dark in light-dark cycle of 30min L-30minD. While the changes could be attributed to NADK2 activity (NADP⁺ synthesis) and NADPP activity (NADP⁺ dephosphorylation).

1. The major deficiency of this study is that it tried to use the data obtained from longer duration (5 – 60 min after onset of illumination, from which the data was collected at 0, 5, 10, 30 and 60 min) to support a hypothesis that, if true, could happen in the first minute after the onset of illumination. The authors seem to ignore the fact that LET can also cause stromal pH change and is the major electron flow after the quick switch from CET to LET shortly after the onset of illumination. Hence, it is illogical to deduct a hypothesis on CET, based on the data obtain in 5, 10, measured after a 25-min illumination. We suggest that **during the first minute of illumination**, the NADP pool becomes rapidly reduced, which induces the association of most of the available FNR with cyt bf, leading to an efficient cyclic electron flow. Induction of CO₂ fixation oxidizes NADPH, inducing a dissociation of the FNR cyt bf complexes and therefore a switch from cyclic to linear flow. In Fig. 4A, cyclic electron flow in an *Ara-*

flow. Curves 4–7 show the kinetics of P700 oxidation measured after 2, 4, 10, and 20 s of dark, following a 15-s pulse of saturating light, sufficient to induce cyclic electron flow (Fig. 4). **The acceleration of the kinetics as a function of the dark time reflects a progressive shift from a cyclic back to a linear mode in the dark ($t_{1/2} \approx 20$ s).** This slow relaxation may reflect a disso-

om a cyclic mode to a liner mode is quick.

2. Lines 53-56. “During this time, although NADP⁺ also does not accumulate in the chloroplasts, it is present at minimal concentrations (ref. 13). It is unclear whether this minimal amount of NADP⁺ is sufficient to drive the LET at the onset of photosynthetic electron transfer.”

In my first review, I have referred the authors to read a paper on NADPH sensor (Lim *et al.*, 2020). However, this article was not cited or discussed in the revised version. Here let me copy a figure from (Lim *et al.*, 2020), in which TKTP-INAP4 is a stromal NADPH sensor, that showed an increase in stromal NADPH after the first min of illumination.

The lag time in stromal NADPH increase (~1 min) could be the time required for the switch from CET (no NADPH production) to LET (NADPH production). There might also be some delay (in seconds) in the confocal operation. Nonetheless, these data showed that, at least at or within one minute after the

onset of illumination, there are sufficient NADP^+ for LET. **If there is really an insufficient NADP^+ at 0 min, and “CET helps regulate NADP pool size via stromal pH regulation” in the first minute, data points at 30sec, 1 min, 2 min, 5 min, instead of 5 min, 10, 30 and 60 min, should be presented.** Note: the kinetics of NADPH decrease in the dark in the above figure is in the same timescale of the data presented in Fig. S14.

3. In addition, the statement of lines 53-56 were based on chloroplast experiments (Fig. S5), in which the NADP^+ level at $t = 0$ is very low. Precaution should be taken as the biochemical of chloroplasts in vitro system is very different from that of intact leaves. Perhaps we could look at the leaf-disc data (Fig. S6), of which the NADP^+ level increased more than 2-fold in 5 min. However, this is quite different from the data published by the authors (Ref. 13). In the following figure, the NADP^+ level only increased $\sim 20\%$ in 5 min of illumination. How different were the experimental conditions from the current manuscript?

Figure 2. Light-dependent kinetics of NADK and NADP.

4. The major data that could support the hypothesis is the data on PGR5 mutant (Fig. 2 and Fig. S6). But again, these data were taken at 5, 10, 30, 60 mins. As point out by the authors (lines 240-241), chloroplasts consume ATP quickly (Ref. 36) and stromal ATP drops to a low level in 30 sec after the light was off (Ref. 36). As a result, NADP^+ synthesis by NADK2 may be hampered quickly by the lack of stromal ATP in the dark. In the light, CET also supplies extra ATP to balance the stromal ATP/ NADPH ratio. The lack of additional ATP generated by the PGR5 pathway could explain why in the *pgr5* mutant, even after prolonged illumination (10-60 min, Fig. S6a), when LET can generate ΔpH , still accumulated less NADP^+ than the WT. This was not seen in the *crr* mutants (Fig. S6c). Could CET contribute an essential amount of stromal ATP for NADP^+ synthesis at the onset of illumination (<1 min), before LET can proceed?

5. In Fig. 2d, isolated chloroplasts treated with AA still exhibited an increase in NADP^+ . What is the implication? The chloroplasts were illuminated for 15-min. LET was dominated and could generate ΔpH to stimulate NADK2 activities and inhibit the NADPP activities, while AA treatment partially reduced, but not completely reduced the NADP^+ accumulation (vs untreated). Could this be explained by the surplus ATP generated by the CET. A balance of the stromal ATP/ NADPH ratio is important for smooth photosynthesis.

In summary, I appreciate the efforts of the authors in producing so much data and explored the effects of pH on NADK and NADPP activities. However, I do not think the hypothesis is watertight, particularly no data points before and after the switch from CET to LET were presented. Therefore, I am afraid I cannot support the publication of this hypothesis.

Minor points:

1. Line 44. “The contribution of PGR5 and NDH pathways to Δ pH formation at 30% and 5%, respectively.” is only based on a pH indicator assay on isolated chloroplasts, whether this percentage hold true in vivo all the time under various illumination condition is under question. This should be specified. More references should be cited too.
2. Line 56. “Photosynthetic electron transfer in PSI is limited without de novo NADP⁺ supply even in low light (Ref.14)”. This is not precise and may mislead reader to think that a lack of NADP⁺, as an PSI electron acceptor, directly affects photosynthetic electron transfer at PSI. In fact, the authors of ref. 14 showed that the translation of PSI A/B, but not of PSII proteins, was downregulated in the *nadk2* mutant. At the same time, the mutant was shown to have lower NADP⁺. The authors proposed a few possibilities how NADK2 affects PSI translation and abundance. This sentence should be rephased to avoid misunderstanding.
3. Fig. 3e. Were the NADPP activities in dark- or light-acclimated chloroplasts different significantly? Please present the statistics.
4. Fig. 3f. There was pH8.0 data point in the graph but it is missing in the figure.
5. Line 164. What is the meaning of “recovered from its delay in *pgr5*”?
6. Line 594. “the stromal pH is relatively favourable for NADPP activity”. What is the stromal pH in the dark? As shown in Fig. 3f, the NADPP is barely active at pH7.5.
7. Line 603. “The reflux of the electron pool in the stroma continues to drive CET for several minutes”. How many minutes? Where is the data to support this? Please cite a suitable reference. Ref. 37 refer to bundle sheath of maize but not Arabidopsis.
8. In the Response 4 of the rebuttal letter the authors also stated that “However, based on our experimental data, the actual amount of NADPH may be larger than that measured by the NADPH sensor”. Can you show me the calculation? Based on what data did you make this conclusion? I am curious.

Reference:

Lim, S.L., Voon, C.P., Guan, X., Yang, Y., Gardestrom, P., and Lim, B.L. (2020). *In planta* study of photosynthesis and photorespiration using NADPH and NADH/NAD⁺ fluorescent protein sensors. *Nature communications* 11 (3238), 3238.

Responses to Reviewer #1

Comment 1:

The major deficiency of this study is that it tried to use the data obtained from longer duration (5 – 60 min after onset of illumination, from which the data was collected at 0, 5, 10, 30 and 60 min) to support a hypothesis that, if true, could happen in the first minute after the onset of illumination. The authors seem to ignore the fact that LET can also cause stromal pH change and is the major electron flow after the quick switch from CET to LET shortly after the onset of illumination. Hence, it is illogical to deduct a hypothesis on CET, based on the data obtain in 5, 10, 30 and 60 mins. I have raised this concern in my last review (Comment 4). However, the reply in the rebuttal letter did not directly answer this shortcoming. Ref. 15. stated that the switch from a cyclic mode to a liner mode is quick ($t_{1/2} = 20$ sec).

Response 1:

The revised version ABSOLUTELY includes data after 1 minute of light exposure in supplementary Fig.12 and Data set. The increase in NADP⁺ and NADPH was slight at 1 min after illumination, supporting our hypothesis that the LET was not fully activated at this stage. Therefore, it did not affect our discussion. I guess the reviewer may see our hypothesis as "ΔpH (pH gradient) produced only by CET dominates NADP⁺ synthesis," but nowhere does it say that. As described in P12Line205, P14Line233-234 and P15Line246-247, the ΔpH via CET acts as a **trigger** that **induces** NADP⁺ synthesis, and the induction is "**delayed**" in the CET mutant and CET inhibition as shown in Fig. 2e, S5, S6 and S12. Our hypothesis indicates that the initial amount of NADP⁺ is low, so the NADP⁺ increase is "delayed" when trying to form ΔpH by LET alone; we do not think or discuss that the switch from CET to LET takes many minutes, and of course we did not state this anywhere. Once NADP⁺ synthesis is induced by CET, the ΔpH formed by LET maintains pH conditions suitable for NADP⁺ synthesis, so our hypothesis is consistent with the reviewer's idea that the ΔpH formed by LET is also important.

Comment 2:

Lines 53-56. "During this time, although NADP⁺ also does not accumulate in the chloroplasts, it is present at minimal concentrations (ref. 13). It is unclear whether this minimal amount of NADP⁺ is sufficient to drive the LET at the onset of photosynthetic electron transfer.". In my first review, I have referred the authors to read a paper on NADPH sensor (Lim et al., 2020). However, this article was not cited or discussed in the revised version. Please pay attention to Fig. 2b of Lim et al., 2020, which showed an increase in stromal NADPH after the first min of illumination.

The lag time in stromal NADPH increase (~1 min) could be the time required for the switch from CET

(no NADPH production) to LET (NADPH production). There might also be some delay (in seconds) in the confocal operation. Nonetheless, these data showed that, at least at or within one minute after the onset of illumination, there are sufficient NADP⁺ for LET. If there is really an insufficient NADP⁺ at 0 min, and “CET helps regulate NADP pool size via stromal pH regulation” in the first minute, data points at 30sec, 1 min, 2 min, 5 min, instead of 5 min, 10, 30 and 60 min, should be presented.

Note: the kinetics of NADPH decrease in the dark in the above figure is in the same timescale of the data presented in Fig. S14.

Response 2:

Of course we read the paper referenced by the reviewer. The paper states that the increase in NADPH saturates in 1 min, but we cannot compare our results with this result for two reasons.

Firstly, we used 3- to 4-week-old rosette leaves, whereas Lim et al. used 10-day-old cotyledons. Although cotyledons also photosynthesize, we cannot equate cotyledons with expanded leaves, which have the greatest potential photosynthetic activity.

Secondly, the reviewer refers the results of an experiment (Fig 2b in Lim et al., 2020) in which the cotyledons were exposed to a high light-intensity of 296 $\mu\text{mol m}^{-2} \text{s}^{-1}$. This light intensity is not at the level of growing light, and so this experimental data shows results under high light-intensity stress conditions. The NADPH detected under these conditions may include the results of NADPH hydrolysis reactions due to the intense light stress response (Maruta et al., 2016).

According to these concerns, we think it difficult to directly compare and did not cite the paper. However, this paper also shows results consistent with ours. Although not mentioned by the reviewer, the same figure (Lim et al., 2020) includes data for 40 $\mu\text{mol m}^{-2} \text{s}^{-1}$: this intensity is relatively weak growing light levels. NADPH continues to increase at least during the 180 sec measurement. This result is consistent with our results, including the data after 1 minute of light exposure, which we added in the revision.

Comment 3:

In addition, the statement of lines 53-56 were based on chloroplast experiments (Fig. S5), in which the NADP⁺ level at $t = 0$ is very low. Precaution should be taken as the biochemical of chloroplasts in *in vitro* system is very different from that of intact leaves. Perhaps we could look at the leaf-disc data (Fig. S6), of which the NADP⁺ level increased more than 2-fold in 5 min. However, this is quite different from the data published by the authors (Ref. 13). In its figure 2e, the NADP⁺ level only increased ~20% in 5 min of illumination. How different were the experimental conditions from the current manuscript?

Response 3:

As described in P5Line73-76 in Introduction and P18Line 304-P19Line316 in Methods, the reason why the increased level of NADP⁺ is different from the previous paper (Hashida et al., 2017) is due to the improvement of measurements. The newly developed method (Ishikawa et al., 2020) was able to eliminate light effects better than the old method, especially in the dark, which improved the quantification of NADP⁺ levels. It is now clear that the increase in NADP⁺ in response to light is actually greater than previously expected.

Comment 4:

4. The major data that could support the hypothesis is the data on PGR5 mutant (Fig. 2 and Fig. S6). But again, these data were taken at 5, 10, 30, 60 mins. As point out by the authors (lines 240-241), chloroplasts consume ATP quickly (Ref. 36) and stromal ATP drops to a low level in 30 sec after the light was off (Ref. 36). As a result, NADP⁺ synthesis by NADK2 may be hampered quickly by the lack of stromal ATP in the dark. In the light, CET also supplies extra ATP to balance the stromal ATP/NADPH ratio. The lack of additional ATP generated by the PGR5 pathway could explain why in the *pgr5* mutant, even after prolonged illumination (10-60 min, Fig. S6a), when LEF can generate Δ pH, still accumulated less NADP⁺ than the WT. This was not seen in the *crr* mutants (Fig. S6c). Could CET contribute an essential amount of stromal ATP for NADP⁺ synthesis at the onset of illumination (<1 min), before LET can proceed?

Response 4:

First, as mentioned in **Response 1 to Comment 1**, the revised version includes data of *pgr5* after 1 minute of light exposure in supplementary Fig.12 and Data set. It is possible that CET supplies the ATP needed for the initial NADP⁺ synthesis. As mentioned in Response3 of previous rebuttal letter, there is currently technically difficult to prove, and our hypothesis does not address this point. We do not consider that all the ATP needed for the NADP pool level to reach a plateau is supplied by CET. Once NADP⁺ synthesis is induced, ATP supplied by LET could be further supplied for subsequent NADP⁺ synthesis. In P14Line237-243, we described the possibility that ATP supply could also be a rate-limiting factor, which is an important topic for future research.

Comment 5:

In Fig. 2d, isolated chloroplasts treated with AA still exhibited an increase in NADP⁺. What is the implication? The chloroplasts were illuminated for 15-min. LET was dominated and could generate Δ pH to stimulate NADK2 activities and inhibit the NADPP activities, while AA treatment partially reduced, but not completely reduced the NADP⁺ accumulation (vs untreated). Could this be explained by the surplus ATP generated by the CET. A balance of the stromal ATP/NADPH ratio is important for smooth photosynthesis.

Response 5:

As described in **Response 1** to **Comment 1**, the ΔpH via CET acts as a **trigger** that **induces** NADP^+ synthesis, and the induction is "**delayed**" in the CET inhibition as shown in Fig. S5. When AA impairs the PGR5 pathway, pH regulation of the stroma will be dependent on LET. Since the initial amount of NADP^+ is low, LET activity is low. This could result in a delay in pH adjustment and consequently a delay in NADP^+ synthesis. Namely, AA impairs PGR5 pathway, resulting in delayed pH conditioning for NADP^+ synthesis.

We proposed a model and discussed our hypothesis based on the data we obtained from our own experiments (Fig. 5). As discussed in Response 3 of the previous rebuttal letter and in **Response 4** above, the current technical difficulties prevent us from proving the involvement of ATP supply regulation in chloroplast NADP^+ synthesis. Therefore, we also described the possibility that ATP supply could also be a rate-limiting factor, which is an important topic for future research because we agree to that a balance of the stromal ATP/NADPH ratio is important for smooth photosynthesis.

Minor points:

Comment 1. Line 44. "The contribution of PGR5 and NDH pathways to ΔpH formation at 30% and 5%, respectively." is only based on a pH indicator assay on isolated chloroplasts, whether this percentage holds true in vivo all the time under various illumination conditions is under question. This should be specified. More references should be cited too.

Response 1. This statement is just a description of what is known about PGR5 and NDH, not significantly related to our results. This is a comment not present in the initial peer review but we will add citations to other relevant literature as needed.

Comment 2. Line 56. "Photosynthetic electron transfer in PSI is limited without de novo NADP^+ supply even in low light (Ref.14)". This is not precise and may mislead reader to think that a lack of NADP^+ , as an PSI electron acceptor, directly affects photosynthetic electron transfer at PSI. In fact, the authors of ref. 14 showed that the translation of PSI A/B, but not of PSII proteins, was downregulated in the *nadk2* mutant. At the same time, the mutant was shown to have lower NADP^+ . The authors proposed a few possibilities how NADK2 affects PSI translation and abundance. This sentence should be rephrased to avoid misunderstanding.

Response 2. As pointed out by the reviewer, our description is not precise because the cited paper does not prove that the reduction of ETR(I) in the *nadk2* mutant is due to less NADP^+ . We agree that it needs to be corrected.

Comment 3. Fig. 3e. Were the NADPP activities in dark- or light-acclimated chloroplasts different significantly? Please present the statistics.

Response 3. Since dark- and light-acclimated chloroplasts have comparable *in vitro* NADPP activity, we think the activity is regulated by stroma pH.

Comment 4. Fig. 3f. There was pH8.0 data point in the graph but it is missing in the figure.

Response 4. Yes. The photo of TLC experiment does not contain pH8.0 data. Instead we demonstrated data of NAD⁺ produced from NADP⁺ using luminescence-based quantification in graph.

Comment 5. Line 164. What is the meaning of “recovered from its delay in *pgr5*”?

Response 5. The increase in NADP pool is delayed in *pgr5* as shown in Fig. 2e, Supplementary Fig. 5 and 12. NaF treatment promoted the increase in NADP pool of *pgr5*, resulting in recovered from delay. The description can be modified for clarity.

Comment 6. Line 594. “the stromal pH is relatively favourable for NADPP activity”. What is the stromal pH in the dark? As shown in Fig. 3f, the NADPP is barely active at pH7.5.

Response 6. Stromal pH in the dark is estimated from pH7.2 to 7.4 (Aranda Sicilia et al., 2021). I will add this information.

Comment 7. Line 603. “The reflux of the electron pool in the stroma continues to drive CET for several minutes”. How many minutes? Where is the data to support this? Please cite a suitable reference. Ref. 37 refer to bundle sheath of maize but not Arabidopsis.

Response 7. Several minutes may be an exaggeration, so we will correct it to “more than one minute. The data in Fig. 4b show that the impairment of P700 oxidation in the dark, an indicator of electron reflux, persists for more than one minute due to the progression of the PGR5 pathway. Others have shown that plastoquinone can be reduced *in vitro* via CET for more than a minute in the presence of NADPH and Fd even in the absence of light in Arabidopsis (Munekage et al., 2002). In the case of NDH pathway, a transient increase of chlorophyll fluorescence after dark is regarded as the cyclic activity by NDH and the transient increase lasts for more than a minute in Arabidopsis (Ishikawa et al., 2008).

Comment 8. In the Response 4 of the rebuttal letter the authors also stated that “However, based on our experimental data, the actual amount of NADPH may be larger than that measured by the NADPH sensor”. Can you show me the calculation? Based on what data did you make this conclusion? I am curious.

Response 8.

My point is that if the NADPH levels under high light conditions in Fig. 2b of Lim et al. (2020), to which the reviewer often refers, are regarded as saturation levels when photosynthesis is most active, an actual saturation levels may be much higher. The data presented in the Lim et al. (2020) are for only three minutes at most after light exposure. Furthermore, the experimental data at $40 \mu\text{mol m}^{-2} \text{s}^{-1}$ did not measure NADPH until saturation, and did not measure NADP^+ . Therefore, the theoretical maximum NADPH level cannot be estimated. Our data reproducibly demonstrated that NADPH does not reach saturation levels in 1 or 5 minutes (Supplementary Fig. 4, 6, 12). From these results, it is not surprising to think that "the actual NADPH saturation level may be higher than the NADPH saturation inferred from the NADPH sensor results".

In summary, all major comments 1-5 by the reviewers have already been addressed in the previous revision. Minor points can be addressed by adding references to comments 1 and 7 and revising the descriptions of comments 2, 5, 6 and 7.

We are grateful to the reviewer#1, the editors and all those involved in brushing up our paper through the peer review process.

Respectfully,
Shin-nosuke Hashida

Reviewer #3 (Remarks to the Author):

Since the beginning of studies on photosynthesis in *Chlorella* (Oh-hama and Miyachi (1960), Masumara (1982)) and in higher plants (Ogren, (1965), Muto et al (1981)), it has been repeatedly observed that a rapid light-induced decrease in the NAD pool is concomitant with an increase in the NADP pool (the reverse occurs after the transition to dark conditions). These observations imply a control of an NAD Kinase (NADK2 in *A. thaliana*) and an NADP phosphatase (still unknown). Despite the importance of this necessary regulatory process, the details have not received much attention since then.

Thus, the manuscript by Ishiyama et al. addresses a very important and neglected aspect of photosynthetic regulation, namely the light-dependent control of NAD⁺ and NADP⁺ pools. This control is important not only for physiologists but also for modelers, because photosynthesis models consider the NAD and NADP pools as constant, which is obviously not the case in vivo. Despite the great potential of this manuscript, all attempts to correlate these changes with the functioning of cyclic electron flow around the PSI are not 100% convincing (see comments below). The correlation between changes in NAD and NADP pools and redox is obvious, but the involvement of CEF is questionable. Therefore, I suggest that the authors propose a more open interpretation of their data

In the following part of this review, I offer some comments that I hope will help authors improve their work.

Main comments

Using sensitive and updated NAD and NADP quantification techniques, Ishiyama et al. reproduce here observations using *Arabidopsis* plants and isolated chloroplasts. They complement their observations with additional biochemical analyses and the use of mutants of cyclic electron flow, envelope transport, ferredoxin/thioredoxin reductase, and thioredoxins. An important result is the demonstration that the transfer of NAD to the NADP pool is a dynamic equilibrium and does not result from a transient activation of plastidial NADK2 in the light (respectively NADP phosphatase in the dark). The authors propose that the pH sensitivity of the two enzymes partly explains the equilibrium of the NAD/NADP pool with the activation of NADK2, respectively the inhibition of NADP phosphatase under alkaline conditions (i.e. light) and the reverse under more acidic conditions (i.e. dark). Other results also suggest a redox control of NADK2.

Redox vs. CET: The timing of activation seems to correlate with activation of cyclic electron flow, but since the latter is also controlled by the redox state of the cells (as proposed for example in the papers by Joliot and Joliot, Johnson and Joliot and Johnson cited in the ms) it is difficult to distinguish between an effect of cyclic or or redox balance.

This is partly due to the experimental approaches:

- Spectroscopy. Measurements of pmf via electrochromic shift at 515-550 nm clearly suffer from the difficulty of correcting for deconvolution of the ECS signal from cytochrome f redox changes (see e.g. Johnson and Ruban 2014, Allorement et al 2018, Wilson et al 2021). Is it possible that the signal measured by the authors in the presence of DCMU and DBMIB is not ECS but rather cytochrome f oxidation, which is maximal in the presence of these inhibitors?

To answer this question, it would be instructive to see the traces of ECS in the presence of DBMIB at the maximum concentration, where the inhibitor should completely collapse the 515-550 nm signal if it is only representative of the proton driving force. Conversely, the signal should be the same (or even greater) if so the difference reflects the oxidation of the cytochrome, which is amplified by the inhibitor).

- Mutants. Using *pgr5* and *ndh* mutants, the authors suggest a greater contribution of the PGR5/PGRL1 pathway in increasing NADP pool size compared to the NDH pathway. However, this observation could also be explained by a redox effect since PSI acceptors are completely reduced in the *pgr5* mutant (as already shown in the original publication by Munekage et al. in 2002, cited in the ms).

In the *kea1kea2* mutant, showing a slow return of stromal pH to more acidic values after dark transfer, the restoration of nocturnal NADP to NAD is also slowed, but this result only confirms the pH-dependence of this regulation, not its dependence on cyclic. The same result was observed with thioredoxin and Fd.Trx reductase mutants, suggesting again a redox control of at least NADK2.

Overall, despite the fact that the activation of NADK2 correlates with the establishment of CET at the dark to light transition, the role of LET/REDOX at steady-state regime is poorly considered. LET (which maintain a redox poise and a pH gradient) could also play the same role as CET to maintain the correct NADP/NAD ratio in the light at steady-state.

By the way, the Redox control of the NADP phosphatase is not analyzed and a straightforward relationship between CAT and the observed changes is not fully validated

Minor points.

The authors should cite the previous works (Oh-hama and Miyachi 1960; Matsumara- kaduta et al. 1982 Ogren, 1965, Muto et al. 1981). describing the light-dependent NAD/NADP pool balance. As results were also obtained with *Chlorella*, the phenomenon they now characterize in more detail is probably general to plants, which an important point.

Line 44 'The contribution of the PGR5 and NDH pathways to Δ pH 45 formation was experimentally estimated at 30% and 5%, respectively'; other publications exist which point to a different contribution (reviewed in Peltier et al below);

Lines 180 to 183 : the consequence of the results are not clearly stated.

Supplementary Fig.7 suggest a redox activation of NADK2: the slow return of the NADP pool to the dark level in thioredoxins mutant is consistent with this observation. However, NADP phosphatase could also be controlled by thioredoxins: inhibition by reduced thioredoxins (light) and activation by oxidized thioredoxins (dark) ? Did the authors test this possibility?

Lines 194-195: check sentence.

Revision of the text in Lines 226 to 243 should be considered. The argumentation is not easy to follow.

Lines 227-229: this is not clear to me. Results in supplementary Fig 2 were obtained with ruptured chloroplasts and with leaf disks in Fig 2F. Do the authors refer to Fig 1F (ruptured isolated chloroplasts) rather than 2F as stated in the text. What do the authors mean by "NADP+ is not fully produced under acidic to neutral pH conditions" and to which figure do they refer? Please clarify.

Statement in line 230-231 seems to be contradictory with text in lines 236-237.

Lines 249-251: "Conversely, the dark-induced decrease in NADP pool size was accelerated in the *pgr5* mutants while *trx-m* and *inap1* showed a delayed decrease in NADP pool size (Fig. 4a, 4c), indicating a positive association with CET activity": I agree for the *pgr5* mutant but for the other mutants results simply indicate that restoration of dark NADP pool size is under the redox control via thioredoxins which are re-oxidized at night. I do not see the connection with CET here.

Lines 557 (Fig 1 e legend): in the absence or presence of NAD +?

Line 62 in Supplementary material: "from" and not "form"

new references quoted in this review.

Oh-hama and Miyachi (1960); <https://doi.org/10.1093/oxfordjournals.pcp.a075762>;
Matsumara- kaduta et al. (1982) BBA, 679, 300-307

Ogren (1965), PMID:4378963,
Muto et al. (1981), <https://doi.org/10.1104/pp.68.2.324>
Peltier et al. (2016). *Annual review of plant biology*, 67, 55-80.
Allorent et al. (2018) *Biochimica et Biophysica Acta (BBA)-Bioenergetics*, 1859(9), 676-683
Wilson et al. (2021). *Plant Physiology*, 187(1), 263-275.
Johnson and Ruban (2014). *Photosynthesis research*, 119(1), 233-242.

Response to Reviewer #3

Comment 1:

Redox vs. CET: The timing of activation seems to correlate with activation of cyclic electron flow, but since the latter is also controlled by the redox state of the cells (as proposed for example in the papers by Joliot and Joliot, Johnson and Joliot and Johnson cited in the ms), it is difficult to distinguish between an effect of cyclic or redox balance. Measurements of *pmf* via electrochromic shift at 515-550 nm clearly suffer from the difficulty of correcting for deconvolution of the ECS signal from cytochrome *f* redox changes (see e.g. Johnson and Ruban 2014, Allorent et al 2018, Wilson et al 2021). Is it possible that the signal measured by the authors in the presence of DCMU and DBMIB is not ECS but rather cytochrome *f* oxidation, which is maximal in the presence of these inhibitors?

To answer this question, it would be instructive to see the traces of ECS in the presence of DBMIB at the maximum concentration, where the inhibitor should completely collapse the 515-550 nm signal if it is only representative of the proton driving force. Conversely, the signal should be the same (or even greater) if so the difference reflects the oxidation of the cytochrome, which is amplified by the inhibitor).

Response 1:

Thank you very much for the helpful comment. In fact, ECS signals are complicated by overlapping absorption changes. If the pulse reflects cytochrome *f* oxidation state as reviewer mentioned, it might be maximal in the presence of DCMU or DBMIB. In our revised manuscript, the total amplitude of the initial trough, that is assumed to be proportional to the total *pmf* (ECS_t) by Klughammer et al., (2013), decreased in dose-dependent manner of DCMU and DBMIB pre-infiltrations (Supplementary Fig.3). Similar to our results, both DCMU and DBMIB application clearly decreased the signal in Cardol et al., (PNAS, 2008). These results suggest that the pulse reflects ECS and the *pmf* is decreased by photochemical inhibitors.

Moreover, we investigated if abolishment of *pmf* suppresses light-responsive NADP increase in revised manuscript (Supplementary Fig.4). In our revised manuscript, we added the data with Nigericin, a monovalent cation-transporting ionophore that acts as an electroneutral antiporter that equilibrates K^+ and H^+ across the membrane, dissipating pH gradient, but preserving membrane potential (Reed 1979). As described in Johnson and Ruban (Photosynth Res 2014), we use 50 μ M Nigericin vacuum-infiltrated wild-type leaves. As expected, Nigericin infiltration clearly suppressed NADP increase, especially NADP⁺. This result strongly suggests an importance of pH gradient across the thylakoid membrane rather than redox state of photochemistry.

Thus, according to the reviewer's helpful comment, we performed additional experiments, and the data undoubtedly strengthened our claim that stromal pH is quite important for adjustment of NADP pool size.

Comment 2:

Using *pgr5* and *ndh* mutants, the authors suggest a greater contribution of the PGR5/PGRL1 pathway in increasing NADP pool size compared to the NDH pathway. However, this observation could also be explained by a redox effect since PSI acceptors are completely reduced in the *pgr5* mutant (as already shown in the original publication

by Munekage et al. in 2002, cited in the ms).

Response 2:

As reviewer pointed out, PSI acceptors are almost completely reduced in *pgr5* mutant under light intensity higher than 100 (Munekage et al., 2002). Importantly, ETR (relative), NPQ and P700 oxidation ratio in *pgr5* at our growth light conditions (less than 100 μmol photons) that are suboptimal for *Arabidopsis* are comparable to those in wildtype. Therefore, we think it is hard to explain our observation by redox effects, since the oxidation state of the PSI acceptor in *pgr5* is comparable to that in wild type under our growth conditions.

However, the reviewer's point is extremely important, raising the possibility that the reducing state of PSI may be involved when NADP pool size is controlled under high light conditions. This is because the redox state of PSI could contribute to further suppression of NADPP activity under high light conditions. We will consider this possibility when advancing our research on the regulation of NADPP activity in future work. In a current, as described in **Response 5** and **Response 9**, we did not observe significant difference in *in vitro* NADP phosphatase activity between light acclimation and dark acclimation (Fig.3e). Thank you very much for your useful comments.

Comment 3:

In the *kea1kea2* mutant, showing a slow return of stromal pH to more acidic values after dark transfer, the restoration of nocturnal NADP to NAD is also slowed, but this result only confirms the pH-dependence of this regulation, not its dependence on cyclic.

Response 3:

In the point of *kea1kea2* mutant, we completely agree with the reviewer's critical comment. As shown in Fig.4, Supplementary Fig13-14 and Supplementary Fig.16, dark responsive NADP pool decrease was accelerated in *pgr5* and in the presence of inhibitor of PGR5-pathway or uncoupler. *kea1kea2* experiment was designed to clarify whether pH is involved in the mechanism by which suppression of the PGR5 pathway and uncoupler treatment accelerated NADP pool decrease; no further results are presented. So, we claim here the importance of stromal pH in controlling dark responsive NADP dynamics and a possible involvement of PGR5-pathway in the pH control. However, as reviewer pointed out, the cyclic pathway is not the only factor regulating stromal pH and outer membrane transporters are also co-regulator of stromal pH. The title of the previous manuscript did not strictly reflect our results and has been changed as follows, "Adjustment of light-responsive NADP dynamics in chloroplasts by stromal pH". As described in Abstract, CET helps regulate NADP pool size via stromal pH regulation, not dominates or directly controls.

Comment 4:

The same result was observed with thioredoxin and Fd-Trx reductase mutants, suggesting again a redox control of at least NADK2.

Response 4:

As reviewer commented, redox-dependent NADK2 regulation might to some extent explain the dynamics of NADP after dark transfer. However, as shown in Supplementary

Fig.2 and Fig.7b, the activation state of NADK2 is not the sole regulator of chloroplastic NADP pool size (please see **Response 13** too). Even when NADK2 activated, pH conditions dominantly regulate NADP production. Basically, we think redox control of NADK2 and pH control of NADP⁺ synthesis are independent of each other. For example, each might regulate a spatially distinct location on the protein. However, its detailed elucidation requires purification of the NADK2 protein, which is currently in progress of another project. In this paper, rather than the biochemical properties of the NADK2 protein, we would like to focus on the fact that phosphorylation of NAD and dephosphorylation of NADP determine the size of the NADP pool, and that the stromal pH is profoundly involved in both the activities. The relationship between Fd-Trx-m pathway and CET is described in **Response 8** and **Response 14** in detail. Anyway, our results demonstrated that NADP pool size is regulated by stromal pH, which could be regulated not only CET but also membrane transporter as described in **Response 3**, so we avoid any statement that misleads readers into thinking that CET is the only factor controlling everything about NADP dynamics.

Comment 5:

Overall, despite the fact that the activation of NADK2 correlates with the establishment of CET at the dark to light transition, the role of LET/REDOX at steady-state regime is poorly considered. LET (which maintains a redox poise and a pH gradient) could also play the same role as CET to maintain the correct NADP/NAD ratio in the light at steady-state. By the way, the Redox control of the NADP phosphatase is not analyzed and a straightforward relationship between CAT and the observed changes is not fully validated.

Response 5:

Thank you very much for your important comment. As the reviewer mentioned, we did not discuss the relationship between NADP pool size and LET/REDOX status at steady-state. Of course, we also recognize the possibility that LET itself contributes to the increase of NADP pool in feedforward loop and to maintain optimal NADP/NAD ratio in the middle of illumination. So, we revised the description explaining Fig.5 in Discussion. However, current knowledge and the materials and techniques available in our laboratory are insufficient to distinguish the extent to which LET, CET, redox, pH gradient, and membrane potential each contribute to NADP dynamics under steady-state light conditions in different leaves. This is because a small change in light condition during an experimental procedure can cause problematic fluctuations in the NADP pool. Therefore, in this study, we conducted experiments under two limited conditions, immediately after light irradiation and immediately after the lights were turned off, focused on the mechanisms at that time.

As the reviewer pointed out, it will be very interesting to see how LET and redox maintains the NADP pool size to a certain level after it reaches the steady state levels (probably after 30-60 min of light exposure). Thank you very much for recognizing this as an important issue for the future of our work.

Honestly, the relationship between NADP phosphatase and CAT has not been analyzed. As reviewer pointed out, current evidence is just the discovery of *in vitro* enzyme activity (Fig. 4c-e), its pH preference (Fig. 4f), and an increase in NADP levels by inhibition of the activity by phosphatase inhibitor (Supplementary Fig. 12). Currently, as described in **Response 2** and **Response 9**, we did not observe significant difference in

in vitro NADP phosphatase activity between light acclimation and dark acclimation (Fig.3e). Moreover, DTT treatment never modulates *in vitro* NADP phosphatase activity (data not shown). We are now in the process of identifying candidate proteins, and the next phase of research will focus on the identification of this molecule. Detailed characterization of enzyme activity and elucidation of the mechanism of NADP dephosphorylation will be discussed in detail in another paper.

Comment 6:

The authors should cite the previous works (Oh-hama and Miyachi 1960; Matsumara-kaduta et al. 1982 Ogren, 1965, Muto et al. 1981), describing the light-dependent NAD/NADP pool balance. As results were also obtained with *Chlorella*, the phenomenon they now characterize in more detail is probably general to plants, which an important point.

Response 6:

Thank you very much for your guidance. We regret not adding these important previous works. We cited these important reports in the revised manuscript.

Comment 7:

Line 44 ‘The contribution of the PGR5 and NDH pathways to Δ pH formation was experimentally estimated at 30% and 5%, respectively’; other publications exist which point to a different contribution (reviewed in Peltier et al below).

Response 7:

Thank you very much for your helpful comment. Our previous manuscript did not adequately describe the difference between PGR pathway and NDH pathway. In our revised manuscript, the functional differences between the two pathways are also described with new citations in Introduction so that it is clear that they are not just redundant pathways.

Comment 8:

Lines 180 to 183: the consequence of the results are not clearly stated.

Response 8:

We apologize that we did not clearly state the result about Fig. 4b and previous reported results. We added some sentences to this paragraph to explain that *trx-m124* mutants and *inap1* have higher CET activity than wild type. *trx-m 124* mutants are deficient in PGR5 downregulation, resulting in high PGR5-dependent CET activity as reported by Okegawa et al., (2022). *inap1* unable to reduce Trx-m protein and has higher CET activity than wildtype. Interestingly, we have previously demonstrated that a light dependent increase in NADP pool is accelerated in *inap1* and *trx-m-124* mutants (Hashida et al., 2018). In the current manuscript, the light-dependent increase in NADP pool is delayed in *pgr5* mutant and in the presence of Antimycin A. Thus, the PGR5-dependent pathway is involved in adjustment of the light-dependent NADP dynamics.

Comment 9:

Supplementary Fig.7 suggest a redox activation of NADK2: the slow return of the NADP

pool to the dark level in thioredoxins mutant is consistent with this observation. However, NADP phosphatase could also be controlled by thioredoxins: inhibition by reduced thioredoxins (light) and activation by oxidized thioredoxins (dark)? Did the authors test this possibility?

Response 9:

Thank you for your insightful comment. As described in **Response 2** and **Response 5**, we did not observe significant difference in *in vitro* NADP phosphatase activity between light acclimation and dark acclimation (Fig.3e). Moreover, DTT treatment never modulates *in vitro* NADP phosphatase activity (data not shown). As shown in Fig. 4f, the difference is observed under various pH conditions. So, we did not set the possibility of redox-dependent NADP phosphatase regulation under our growth conditions. However, current knowledge does not completely exclude possible involvement in regulation under high light conditions as described in **Response 2**. We will investigate the possibility after molecular identification of NADP phosphatase in future work.

Comment 10:

Lines 194-195: check sentence.

Response 10:

We have split the sentence into two sentences as follows, “*Two envelope transporters, KEA1 and KEA2, adjust stromal pH during light to dark transition. Importantly, the rate of neutralization of stromal pH is significantly impaired in kea1kea2 mutants after dark transition*”.

Comment 11:

Revision of the text in Lines 226 to 243 should be considered. The argumentation is not easy to follow.

Response 11:

Thank you very much for the suggestion. Together with corrections to points raised in other comments, we have revised the paragraph to make it easier to read and understand. Please see **Response 12** and **Response 13** too.

Comment 12:

Lines 227-229: this is not clear to me. Results in supplementary Fig 2 were obtained with ruptured chloroplasts and with leaf disks in Fig 2F. Do the authors refer to Fig 1F (ruptured isolated chloroplasts) rather than 2F as stated in the text. What do the authors mean by “NADP+ is not fully produced under acidic to neutral pH conditions” and to which figure do they refer? Please clarify.

Response 12:

We are very sorry for the trouble we have caused you. This is absolutely our mistake for Figure citation. We should refer to Fig.1f and supplementary Fig.2 instead of Fig.2f. We made a correction.

Comment 13:

Statement in line 230-231 seems to be contradictory with text in lines 236-237.

Response 13:

Thank you for pointing out the ambiguity in our description. As the reviewer commented, these two sentences can also be read to be contradictory. What we wanted to explain in the first sentence is that the activation state of NADK2 is not the sole regulator of chloroplastic NADP pool size. We have revised it as follows, “*In fact, the in vitro NADP-producing activity was >10-fold higher than that observed in organello, suggesting that the activation state of NADK2 is not the only regulator of chloroplastic NADP pool size (Fig. 1e and f)*”.

And, what we wanted to explain in the second sentence is that the rate of NADP increase is regulated by the balance between NAD phosphorylating activity and NADP dephosphorylating activity at the beginning of illumination. We have revised it as follows, “*Thus, a rate of NADP pool size increase was adjusted through the balance between NAD⁺ phosphorylating activity and NADP⁺ dephosphorylating activity at the beginning of illumination*”.

Comment 14:

Lines 249-251: “Conversely, the dark-induced decrease in NADP pool size was accelerated in the *pgr5* mutants while *trx-m* and *inap1* showed a delayed decrease in NADP pool size (Fig. 4a, 4c), indicating a positive association with CET activity”: I agree for the *pgr5* mutant but for the other mutants results simply indicate that restoration of dark NADP pool size is under the redox control via thioredoxins which are re-oxidized at night. I do not see the connection with CET here.

Response 14:

Thank you for pointing out our careless description. We have revised the sentence to clarify the relationship to CET as follows, “*Conversely, the dark-induced decrease in NADP pool size was accelerated in the pgr5 mutants while that was decelerated in the mutants, trx-m124 and inap1, with high PGR5-dependent CET activity and pmf (Fig. 4a–d), indicating that higher CET activity is more likely to increase NADP pool size and less likely to decrease NADP pool size*”.

It has been reported that Fd-Trx-m negatively controls PGR5-dependent cycling pathway according to Okegawa et al., 2021 we have cited. We also confirmed that *inap1* (Trx-m reductase mutant) and *trx-m* triple mutants have slow P700 oxidation rate induced by far-red light (Fig. 4b). These results suggest a defect in the redox-dependent suppression mechanism of PGR5-cyclic pathway in *inap1* and *trx-m* triple mutants, meaning PGR5 pathway is high in *inap1* and *trx-m* triple mutants. After dark transfer, the stromal electron pool is back to plastoquinone by CET activity and the electron is transferred to PSI again until the stromal electron pool is depleted.

Comment 15:

Lines 557 (Fig 1 e legend): in the absence or presence of NAD +?

Response 15:

Thank you very much for correcting the mistake.

Comment 16:

Line 62 in Supplementary material: “from” and not “form”

Response 16:

Thank you very much for correcting the mistake.

We are grateful to the reviewer#3, the editors and all those involved in brushing up our paper through the peer review process.

Respectfully,
Shin-nosuke Hashida

Reviewer #4 (Remarks to the Author):

This is a very interesting study on a highly topical subject – regulation of the relative content of NADH&NAD⁺ and NADPH&NADP⁺ in chloroplasts. The authors have made extensive measurements to monitor the contents of these components in a range of low light conditions and dark treatments, in wild type Arabidopsis treated with inhibitors considered specific to various locations in the photosynthetic electron transport chain and in a set of Arabidopsis mutants lacking proteins important for cyclic electron transport and ion fluxes across the thylakoid membrane.

This is my first sight of the manuscript, which includes a response to a single previous review.

Their data suggest differences in the balance between the two phosphorylated and dephosphorylated forms based on light, inhibitor and genetic studies. However, it is hard to judge how significant these differences are, as there is a lack of statistical analysis throughout. I find one reference to a Tukey test (line 163), but it is unclear what comparison is being made (initial light treatments, or time in the dark). Often the authors describe rather small relative changes, and without statistics it is difficult to judge whether they are meaningful.

Moreover, there are several things in the data that I find very hard to understand, possibly as the legends are extremely brief. In many cases I was not clear whether experiments had been performed on leaf discs or on isolated chloroplasts (e.g. the whole of Figure 4). My main points of confusion were:

Why have the authors used such low light intensities? These represent deep, deep shade, and in some cases are below the compensation point

What do the black bars above figure represent? Probably the dark for the solid black part, but the single line in e.g. Figure 1a?

Were the inhibitors added in solvent or water? This can have a big effect!

Were the control leaf discs vacuum infiltrated? If so, with what?

What is the difference between Figure 2a and figure 3c? what is the purpose of figure 3c?

Do figure 2a and 2d represent different experiments? How many replicates? Why is the relationship between dark and control so different between them? The NADP is so much lower in 2d.

How were the lines drawn on the graph in Figure 3a? I'm not sure how the slopes were calculated, but on the graph there are no points on the 70 $\mu\text{mol m}^{-2} \text{s}^{-1}$ gradient, which is problematic.

What is the graph to the right of panel f in Figure 2? How was this calculated?

Why are there no data points on Figure 4a c and f?

Why is the wt line so different between Figure 4a, c and f? This difference is as great as any variation between genotypes. Why do the antimycin A and NH₄ in 4f not phenocopy the pgr5 mutant? They should if the lack of ΔpH is accelerating the phosphatase reaction.

Why do the crr mutant traces in Figure 4a show the opposite trend in Supplementary Figure 14, which seems to be the same measurement?

In addition, there are major concerns about interpretation.

1. I am not sure the emphasis on cyclic during light induction is appropriate. DCMU, which should allow cyclic to proceed, has the greatest effect on NADPH&NADP⁺ accumulation of any inhibitor or mutation.

2. The emphasis on pH may also be misguided. Stromal pH does fluctuate, but not by so much – maybe from pH 7 to pH 7.8 at most, and recent fluorescent probes indicate a lot less (7.32 to 7.55 was reported by Luu Trinh & Masuda in 2022). In this context, many of the pH measurements reported in this paper are at unphysiological pH, and the change in activity may not be so great in the chloroplast. Indeed, at such low light intensities ΔpH would dissipate quite rapidly, in a few minutes at the longest, while changes in NADP contents presented here occur over 10-20 minutes

3. Cyclic electron transfer implies return of electrons to the plastoquinone pool after their excitation at PSI. In the dark, this process will stop, so for impact on NADP dephosphorylation the authors are not really talking about a difference in cyclic electron transport, but it's secondary effects. The impact seen in the dark in the cyclic mutants in Supplementary Figure 14 (and Figure 4a for pgr5 only) may be related to the relative reduction state of the stroma (which would in turn impact on thiol regulation of the kinase) or could be related to ion fluxes in the dark.

I wish the authors well in publishing their work. They are on to something very interesting, but at present, from what I understand of the data, I do not think the conclusions are justified.

Response to Reviewer #4

Comment 1:

Their data suggest differences in the balance between the two phosphorylated and dephosphorylated forms based on light, inhibitor and genetic studies. However, it is hard to judge how significant these differences are, as there is a lack of statistical analysis throughout. I find one reference to a Tukey test (line 163), but it is unclear what comparison is being made (initial light treatments, or time in the dark). Often the authors describe rather small relative changes, and without statistics it is difficult to judge whether they are meaningful. Moreover, there are several things in the data that I find very hard to understand, possibly as the legends are extremely brief. In many cases I was not clear whether experiments had been performed on leaf discs or on isolated chloroplasts (e.g. the whole of Figure 4).

Response 1:

Thank you for your comment. We believe that statistical analysis is a really important too and added description about the statistical test in results section and figure legends. In addition, details were added to the figure legends to the extent that they do not duplicate the methods section.

Comment 2:

Why have the authors used such low light intensities? These represent deep, deep shade, and in some cases are below the compensation point.

Response 2:

Under natural conditions, light gradually becomes more intense as night turns to morning. Photosynthesis is induced by light. In this paper, we focus primarily on the increase in the NADP pool during the photosynthesis induction phase of dark-acclimated leaves. Based on the fact, we do not believe that the light intensities in our experiment are too low. Furthermore, Arabidopsis can grow well and normally expand true leaves.

Comment 3:

What do the black bars above figure represent? Probably the dark for the solid black part, but the single line in e.g. Figure 1a?

Response 3:

Sorry for the confusion. We have added the missing explanation in figure legends. The vertical line in Figure 1c shows the switching point of light intensity, but I removed the vertical line because it is also shown as triangles.

Comment 4:

Were the inhibitors added in solvent or water? This can have a big effect! Were the control leaf discs vacuum infiltrated? If so, with what?

Response 4:

Of course, the inhibitor is dissolved in a solvent, so the control also uses a solvent. We added "As a control, leaves were treated with 1% (v/v) ethanol (used as a solvent for the inhibitors)" to the method.

Comment 5:

What is the difference between Figure 2a and figure 3c? what is the purpose of figure 3c?

Response 5:

These two experiments have distinctly different purposes. In Figure 2a, the DCMU was treated on dark-acclimated leaves that has minimum NADP pool size. In Figure 3c, the DCMU was treated on light-acclimated leaves that has maximum NADP pool size. These two experiments evaluate the effect of DCMU on increasing the NADP pool and the effect of DCMU on decreasing the NADP pool, respectively. As noted in the previous version of the method, infiltration was not performed during the treatment of light-acclimated leaves in order to reduce the effects of light intensity fluctuations during experimental procedures. To make this clear, we have also added this description in figure legends.

Comment 6:

Do figure 2a and 2d represent different experiments? How many replicates? Why is the relationship between dark and control so different between them? The NADP is so much lower in 2d.

Response 6:

Leaf disc data is shown in Fig. 2a and isolated chloroplasts data is shown in Fig. 2d. We guess that this comment is referring to the difference between Fig 2c and Fig 2d. The number of repetitions in both of these experiments is 6, but the timing of the experiments conducted in Fig. 2c and Fig. 2d is different. Different researchers isolated chloroplasts at different times of the year, and this has led to variations in their quality, resulting in large differences in the dark and control values between Fig 2c and Fig 2d. We never compare data between different experiments. We only evaluate differences between treatments within the same time period when the quality of the isolated chloroplasts is relatively consistent. Both experiments reproduced the phenomenon of light-induced increase in the NADP pool.

Comment 7:

How were the lines drawn on the graph in Figure 3a? I'm not sure how the slopes were calculated,

but on the graph there are no points on the 70 $\mu\text{mol m}^{-2} \text{s}^{-1}$ gradient, which is problematic.

Response 7:

This is curve fitting in ImageJ software as described in the method of previous version. The dark condition reduces the NADP pool from a maximum value to a minimum value. The time it takes to decrease by half is calculated from the regression equation of the curve. Thanks to the reviewer's suggestion, we realized that this figure contained a serious error in which the curve did not match the raw data plot. We sincerely appreciate it. It appears that when the plot and the curve were overlaid, the points on plot were moved. We have corrected the error. Thank you very much for pointing this out.

Comment 8:

What is the graph to the right of panel f in Figure 2? How was this calculated?

Response 8:

We guess reviewer mentioned about Figure 3f. Thank you for pointing this out. I apologize for the confusing illustration; the graph to the right of f should have been independent as g. The graph is NAD⁺ production activity from NADP⁺ by ruptured chloroplasts. As described in the NADP⁺ dephosphorylation assay section in methods of previous version, luminescence-based assay was used to detect NAD⁺ dephosphorylated from NADP⁺ to NAD⁺.

Comment 9:

Why are there no data points on Figure 4a c and f?

Response 9:

All plot data for Figures 4a, c and f are shown in Supplementary figures 13 and 14. Due to the large amount of information, only the curves are shown here to avoid a complicated figure.

Comment 10:

Why is the wt line so different between Figure 4a, c and f? This difference is as great as any variation between genotypes.

Response 10

As reviewers know, environmental conditions cause various variability in plants. In our plant cultivation room in Japan, we try to maintain the same conditions as much as possible throughout the year, but there are inevitably seasonal differences in temperature and humidity. Therefore, even when grown under the same cultivation procedure, there are differences in leaf thickness and chlorophyll content at different times of the year. For this reason, when conducting comparative experiments between genotypes, only individuals grown at the same time of year should be used in the experiments. Of course, the results measured at different times of the year

are not mixed and analyzed, so they are shown as independent graphs. This is the reason why WT varies slightly from graph to graph. The fact that some differences in the dynamics of the NADP pool are observed in different seasons implies that there are unexplored controlling factors. The hypothesis we propose in this study is just a first step toward a complete picture. We look forward to further progress in this research area.

Comment 11:

Why do the antimycin A and NH₄ in 4f not phenocopy the *pgr5* mutant? They should if the lack of delta pH is accelerating the phosphatase reaction.

Response 11:

As reviewer points out, it would be great if inhibitors could completely phenocopy *pgr5*. However, we do not believe that *pgr5* can be completely phenocopied by inhibitors. This is because inhibitors do not specifically inhibit only the activity of PGR5. Moreover, we think *pgr5* mutants are affected secondarily by the reduced activity of CET during growth.

Comment 12:

Why do the *crr* mutant traces in Figure 4a show the opposite trend in Supplementary Figure 14, which seems to be the same measurement?

Response 12:

That is because Figure 4a and Supplementary figure 13 shows the NADP pool (sum of NADP⁺ and NADPH) while Supplemental figure 14 shows NADPH only. This indicates that the difference in the dynamics of NADP pool decrease is mainly due to the dephosphorylation of NADP⁺ shown in Supplemental Figure 13.

Comment 13:

I am not sure the emphasis on cyclic during light induction is appropriate. DCMU, which should allow cyclic to proceed, has the greatest effect on NADPH&NADP⁺ accumulation of any inhibitor or mutation.

Response 13:

As the reviewer pointed out, DCMU does not directly inhibit CET. As well as reviewer, we believe that the involvement of CET in the photosynthesis induction phase is essentially just a trigger in dark-acclimated leaves too. Importantly, DBMIB treatment also suppressed the increase in the NADP pool during photosynthesis induction (Fig. 2a and 2b), suggesting that CET contributes to the increase in the NADP pool. Nevertheless, the discussion has been revised to weaken the wording as follows to avoid overemphasis.

P11 Line16-17 “The present study suggests a potential role of CET in regulating de novo NADP⁺

supply as the electron acceptor of LET”.

P13 Line1-3 " A hypothetical model shown in Fig. 5 explains the possible mechanism of NADP homeostasis during photosynthesis under fluctuating natural light conditions".

Comment 14:

The emphasis on pH may also be misguided. Stromal pH does fluctuate, but not by so much – maybe from pH 7 to pH 7.8 at most, and recent fluorescent probes indicate a lot less (7.32 to 7.55 was reported by Luu Trinh & Masuda in 2022). In this context, many of the pH measurements reported in this paper are at unphysiological pH, and the change in activity may not be so great in the chloroplast. Indeed, at such low light intensities Δ pH would dissipate quite rapidly, in a few minutes at the longest, while changes in NADP contents presented here occur over 10-20 minutes.

Response 14:

We agree with the reviewer's point about pH change. We consider that stroma pH fluctuations are not dramatic and are in the range of pH 7-7.5 from recent studies. The changes in enzyme activity related to NADP pool size detected across this pH range are not very small. In Fig. 3f and 3g, the NADP⁺ dephosphorylation activity changes dramatically from pH 7 to pH 7.5, and the variation of activity at pH 7.2 is very large. These results suggest NADP⁺ dephosphorylation activity is dramatically regulated in this pH range. In contrast, NAD⁺ phosphorylation activity is greatly elevated at pH 7-7.5. We cannot ignore the possibility that both activities are inversely regulated by pH to achieve synergistic activity changes in NADP pool size regulation. We are now conducting a detailed biochemical analysis of the protein responsible for these activities, however, so far, a single fusion protein purified in *E. coli* cannot reproduce the expected enzymatic properties. We would like to investigate the possibility that the activity is regulated by a multi-protein complex. We also do not intend to misguide readers that stromal pH dominates NADP pool size. In fact, we use “Adjustment” in the title, not “Control” or “Regulation”.

Comment 15:

Cyclic electron transfer implies return of electrons to the plastoquinone pool after their excitation at PSI. In the dark, this process will stop, so for impact on NADP dephosphorylation the authors are not really talking about a difference in cyclic electron transport, but it's secondary effects. The impact seen in the dark in the cyclic mutants in Supplementary Figure 14 (and Figure 4a for pgr5 only) may be related to the relative reduction state of the stroma (which would in turn impact on thiol regulation of the kinase) or could be related to ion fluxes in the dark.

Response 15:

Thank you for your very important comment. As reviewer said, CET is the reflux of electrons

excited by PSI in strict definition. Electrons excited from PSI return to the plastoquinone pool via ferredoxin or thiols from other proteins. Of course, the phenomenon we are detecting in the dark is the return of the reducing power present in the stroma to the plastoquinone pool through ferredoxin or thiols. This flow shares the CET pathway, so CET activity is necessary. As the reviewer points out, electron reflux in the dark is not strictly called CET, but I think our data indicate that CET activity by PGR5 is important. Although this study shows that stroma pH is involved in NADP pool size regulation, it is not our intention to exclude other possibilities exemplified by the reviewer. We have added the following sentence to the discussion with new citations.

P13 Line1-14 "Besides electron reflux by CET, processes that consume reducing power in the stroma in the dark and ion flux activity may be involved in regulating NADP pool sizes^{45, 46}, and there will be an integrated control system that includes these processes."

We are really grateful to the reviewer#4, the editors and all those involved in brushing up our paper through the peer review process.

Respectfully,

Shin-nosuke Hashida

Reviewer #4 (Remarks to the Author):

The authors have made their paper much clearer and easier to follow. Greater understanding of their experiments makes it easy to understand what has been done. I think this work is an important contribution to the field.

It is clear that there is a strong correlation between delta pH generation and phosphorylation of the NAD pool to the NADP pool. This is now reflected in the title, and is an important finding. However, the possibility that CET is a directly regulator of this activity is still heavily over-emphasized throughout the manuscript, especially the end of the abstract and final section of the discussion, and I do not think the data presented are strong enough to support this. I do not mean that more data are needed, I mean that the data show something different. The CET mutants and inhibitors do tell the authors useful things about how delta pH is controlling interconversion of NAD and NADP pools, but they do not show a specific role for CET per se.

Clearly, CET contributes to delta pH, and the *pgr5* mutants were initially isolated based on their poor generation of delta pH, so the poor NAD > NADP conversion supports this. However, in steady state conditions LET makes the major contribution to delta pH, and therefore is even more important for conversion of the NAD pool to NADP. Two examples in the manuscript where slightly misinterpreted data imply a greater role for CET than is likely include the following:

1. The most convincing proof of a direct role for CET comes from incubation of isolated chloroplasts in antimycin A, which strongly inhibits NADP pool accumulation (Fig 2D). However, most preparations of Arabidopsis chloroplasts have poor functionality in LET, and therefore conduct a lot of CET by default, which means that in this experiment, antimycin A inhibits a much greater proportion of electron transport than it would in leaves (where the effect is much less e.g. Figure 4F). It may be that the authors were able to make chloroplasts from Arabidopsis capable of LET but this is hard to judge without O₂ electrode data to prove this. In the absence of such proof, I would interpret this data as showing that in conditions where CET is performing a greater proportion of total delta pH generation, inhibiting CET has more of a negative impact on NAD > NADP conversion.

2. in Fig 4A, the rapid (relative) decrease in NADPH seen in *pgr5* in the dark is almost certainly because the phosphatase was not deactivated in the first place (due to low delta pH in *pgr5*), and there is therefore little inhibitory pmf to relax. It seems very unlikely from the data here that CET in the dark is involved. I would interpret this result as showing poor light deactivation of the phosphatase in *pgr5*, due to low delta pH, results in rapid reversion of the NAD and NADP pools, but that this is a function of the activation state of the phosphatase in the prior light treatment.

In short, if the authors simplify their message to say that delta pH regulates interconversion of the NAD and NADP pools, they have a strong paper worth publishing. Based on the data presented, over-emphasizing a role for cyclic puts them in danger of misleading the community.

Response to Reviewer #4

Comment 1:

The authors have made their paper much clearer and easier to follow. Greater understanding of their experiments makes it easy to understand what has been done. I think this work is an important contribution to the field.

Response 1:

We are pleased to receive these comments. Our paper has been revised in accordance with your peer review comments. Thank you very much for the peer review.

Comment 2:

It is clear that there is a strong correlation between delta pH generation and phosphorylation of the NAD pool to the NADP pool. This is now reflected in the title, and is an important finding. However, the possibility that CET is a directly regulator of this activity is still heavily over-emphasized throughout the manuscript, especially the end of the abstract and final section of the discussion, and I do not think the data presented are strong enough to support this. I do not mean that more data are needed, I mean that the data show something different. The CET mutants and inhibitors do tell the authors useful things about how delta pH is controlling interconversion of NAD and NADP pools, but they do not show a specific role for CET per se.

Response 2:

Thanks for pointing this out. We too do not believe that CET directly controls the interconversion of the NAD and NADP pools, so we do not intend to mislead. We have revised the excessive emphasis throughout the manuscript.

Comment 3:

Clearly, CET contributes to delta pH, and the *pgr5* mutants were initially isolated based on their poor generation of delta pH, so the poor NAD > NADP conversion supports this. However, in steady state conditions LET makes the major contribution to delta pH, and therefore is even more important for conversion of the NAD pool to NADP. Two examples in the manuscript where slightly misinterpreted data imply a greater role for CET than is likely include the following:

1. The most convincing proof of a direct role for CET comes from incubation of isolated chloroplasts in antimycin A, which strongly inhibits NADP pool accumulation (Fig 2D). However, most preparations of Arabidopsis chloroplasts have poor functionality in LET, and therefore conduct a lot of CET by default, which means that in this experiment, antimycin A inhibits a much greater proportion of electron transport than it would in leaves (where the effect is much less e.g. Figure 4F). It may be that the authors were able to make chloroplasts from Arabidopsis capable of LET but this is hard to

judge without O₂ electrode data to prove this. In the absence of such proof, I would interpret this data as showing that in conditions where CET is performing a greater proportion of total delta pH generation, inhibiting CET has more of a negative impact on NAD > NADP conversion.

Response 3:

As the reviewer points out, our data from isolated chloroplast experiments do not provide the driving ratio of CET to LET, so we do not know whether CET regulates NAD to NADP conversion or is involved in the regulation. We have toned it down throughout the manuscript. The statement meaning "CET directly regulates *de novo* NADP⁺ supply" has been revised to mean "CET is involved in the regulation of *de novo* NADP⁺ supply." We are very grateful to the reviewer for pointing this out as it is an extremely important point in our study.

Comment 4:

2. in Fig 4A, the rapid (relative) decrease in NADPH seen in *pgr5* in the dark is almost certainly because the phosphatase was not deactivated in the first place (due to low delta pH in *pgr5*), and there is therefore little inhibitory pmf to relax. It seems very unlikely from the data here that CET in the dark is involved. I would interpret this result as showing poor light deactivation of the phosphatase in *pgr5*, due to low delta pH, results in rapid reconversion of the NAD and NADP pools, but that this is a function of the activation state of the phosphatase in the prior light treatment.

Response 4:

Our data include NADP⁺, NADPH, and NADP pools. This may be confusing, but the data in Fig. 4a shows the NADP pool and the NADPH pool is shown in Supplementary Fig14.

As the reviewer mentioned, it is also possible that the NADP phosphatase (NADPP) is not sufficiently inactivated by light in *pgr5*. As a result, NADPP may retain high activity in *pgr5* even in the light and may quickly reduce the NADP pool upon shading. An important point is that in the wild type, the electron pool in the stroma is refluxed via CET even during light shading. This is the measurement principle of CET. In our discussion, we point out that the balance of NAD and NADP interconversion maintained by this brief Δ pH formation may be disturbed by the loss of CET activity in *pgr5*, resulting in a relatively rapid decrease in the NADP pool. We believe this discussion is essentially consistent with the reviewer's considerations. Of course, identification and detailed biochemical analysis of NADPP are needed to determine the extent to which this change in activity balance contributes to the size of the NADP pool in our future work.

As in **Response 3**, our data are insufficient to conclude that "CET regulates NADP status upon shading". Therefore, we have revised the manuscript to avoid overemphasizing the role of CET.

Comment 5:

In short, if the authors simplify their message to say that delta pH regulates interconversion of the

NAD and NADP pools, they have a strong paper worth publishing. Based on the data presented, over-emphasizing a role for cyclic puts them in danger of misleading the community.

Response 5:

Thank you very much for your comments, and I have revised the entire manuscript to agree that CET is important as a ΔpH -forming factor, but that it is ΔpH and not CET itself that regulates NADP pool size.

We are really grateful to the reviewer#4, the editors and all those involved in brushing up our paper through the peer review process.

Respectfully,

Shin-nosuke Hashida